# DreamSteerer: Enhancing Source Image Conditioned Editability using Personalized Diffusion Models

**Zhengyang Yu**[1]*        **Zhaoyuan Yang**[2]        **Jing Zhang**[1]†

Australian National University[1]    GE Research[2]

{zhengyang.yu,jing.zhang}@anu.edu.au    zhaoyuan.yang@ge.com

## Abstract

Recent text-to-image personalization methods have shown great promise in teaching a diffusion model user-specified concepts given a few images for reusing the acquired concepts in a novel context. With massive efforts being dedicated to personalized generation, a promising extension is personalized editing, namely to edit an image using personalized concepts, which can provide a more precise guidance signal than traditional textual guidance. To address this, a straightforward solution is to incorporate a personalized diffusion model with a text-driven editing framework. However, such a solution often shows unsatisfactory editability on the source image. To address this, we propose DreamSteerer, a plug-in method for augmenting existing T2I personalization methods. Specifically, we enhance the source image conditioned editability of a personalized diffusion model via a novel Editability Driven Score Distillation (EDSD) objective. Moreover, we identify a mode trapping issue with EDSD, and propose a mode shifting regularization with spatial feature guided sampling to avoid such an issue. We further employ two key modifications to the Delta Denoising Score framework that enable high-fidelity local editing with personalized concepts. Extensive experiments validate that DreamSteerer can significantly improve the editability of several T2I personalization baselines while being computationally efficient. Project page: https://github.com/Dijkstra14/DreamSteerer.

## 1  Introduction

Text-to-Image Diffusion Probabilistic Models (T2I DPMs) [58, 61] have revolutionized novel content creation due to their superior capacity in both sample fidelity and mode coverage, as well as their flexibility in achieving effortless concept composition [44] and user-friendly control [35]. Despite the success of these models, the limited expressiveness of natural language may lead to ambiguity in real-world scenarios where users demand greater specificity and engagement in the creation process.

This challenge has sparked extensive research in *T2I Personalization* [10, 16, 62]. Specifically, personalization is the process of teaching T2I DPMs a novel visual concept with a few (usually 3-8) reference images by fine-tuning the pre-trained model parameters. After personalization, the personal concept is linked to a rare token in the text encoder dictionary, e.g., "[my_dog]", enabling flexible reuse of the visual concept in new contexts, e.g., "a photo of [my_dog] wearing an astronaut suit". Despite the success of these models, the majority of previous works have been focusing on personalized generation, emphasizing the preservation of concept identity across varying textual conditions. In this work, we explore a natural extension on these methods to perform image editing with acquired concepts, namely *personalized editing*, which can be promising in enabling higher level

---

*Work was partially done while Zhengyang Yu was an intern at GE Research.

†Corresponding author.

38th Conference on Neural Information Processing Systems (NeurIPS 2024).

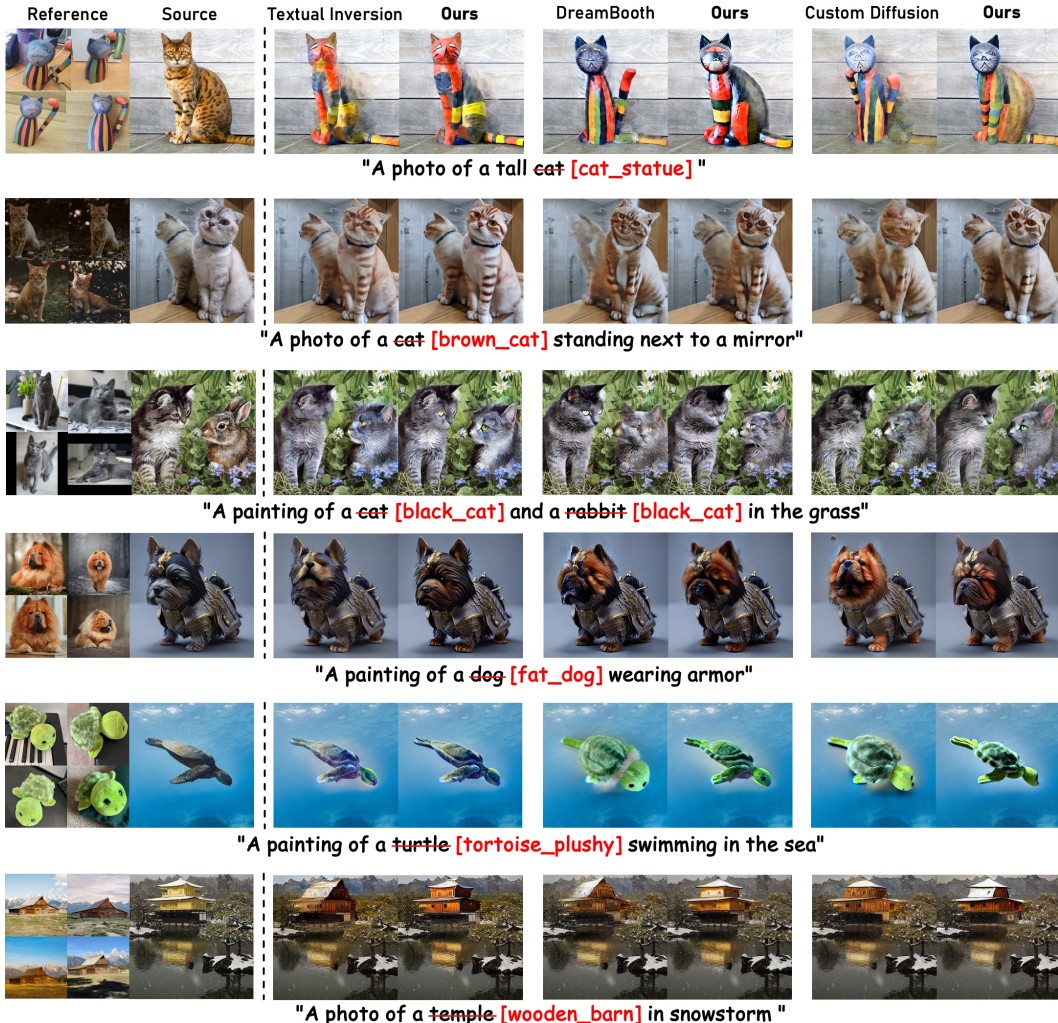

Figure 1: DreamSteerer enables efficient editability enhancement for a source image with any existing T2I personalization models, leading to significantly improved editing fidelity in various challenging scenarios. When the structural difference between source and reference images are significant, it can naturally adapt to the source while maintaining the appearance learned from the personal concept.

control over the editing direction than traditional text-driven editing frameworks [21, 34, 55]. The primary goal is to synthesize a high-fidelity image that aligns the appearance and content of the target concept, as well as the structural layout and background of the source image without blindly resorting to copying learned reference images. A straightforward solution to this problem is to incorporate a personalized diffusion model into an existing text-driven editing model, by which personalized editing appears to be trivial via swapping a source subject token in the source prompt by the special token, e.g., "a photo of (dog → [my_dog]) wearing an astronaut suit". However, such a naïve incorporation often leads to severe distortion or failure in natural adaptation to the source image layout. We attribute the essential causes for such failure to the limited scope of reference images in personalization. Under a lack of data diversity, existing personalization methods are prone to collapsing into the patterns of reference images [71] and entangling subject-relevant and subject-irrelevant information [9]. This causes a significant loss in the model's prior knowledge related to the source category, resulting in poor editability [48, 60] in a new context. Additionally, the stricter demands for maintaining structural layout [54, 72] and preserving subject-irrelevant information [12, 21] necessitate a higher level of editability than personalized generation. In more challenging editing scenarios, a certain level of extrapolation on the attributes of personalized concepts may be required, e.g. the editing process may require significant change in subject structure, pose or style to achieve a high-fidelity

editing result on the source image. Without information on the source image during personalization, existing personalization methods can hardly be guaranteed to be editable on arbitrary source images.

To overcome these challenges, in this paper, we propose the first approach that enhances the source image conditioned editability using existing personalized T2I DPMs (DreamSteerer). Inspired by the one-step Bayes optimal denoising using Tweedie's formula [15, 59] with DPMs [3, 7, 11, 47, 69], and the probabilistic score distillation sampling [22, 57, 74], we formulate editability enhancement on the source image as a novel score distillation objective, dubbed Editability Driven Score Distillation (EDSD). Then we identify the existence of a mode trapping issue when directly optimizing EDSD. Inspired by recent success in zero-shot semantic correspondence [1, 6, 80] based on the strong spatial awareness of UNet attention features, we propose a spatial guided sampling strategy that produces a regularization set which alleviates the mode trapping issue via shifting the mode of personalized DPMs distribution. We further make two major modifications on the Delta Denoising Score (DDS) [22] framework for a valid adaptation from text-driven editing to personalized editing.

We summarize our main contributions as: 1) we identify and analyze the lack of editability in existing T2I personalization methods for editing real-world images, 2) we propose the DreamSteerer framework, a plug-in method that is compatible with arbitrary personalization baselines and requires only a small number of fine-tuning steps ($\sim 10$) to achieve significant improvement in editability on a source image, 3) we validate the effectiveness of DreamSteerer on 3 different baselines [16, 39, 62], demonstrating its efficacy, especially in challenging personalized editing scenarios such as significant structural gaps and data-hungry cases (Fig. 1). See Fig. 3 for an overall framework.

## 2 Related Work

**T2I Personalization.**  Building upon the success of T2I DPMs [52, 61, 64], recent works have shown the promise of personalized T2I synthesis [16, 19, 39, 62]. Textual Inversion [16] encapsulates personalized concepts by word embedding optimization, which is further improved by more expressive representation spaces [2, 73]. DreamBooth [62] fine-tunes the full model parameters of a pre-trained DPM that conditions on a rare word token. Efficient fine-tuning methods [9, 18, 45, 71, 77] have been introduced for improving efficiency and generalizability. Custom Diffusion [10] optimizes both a word embedding as well as the UNet cross-attention key and value projections for improved compositionality. We explore enhancement in editability on these three different types of models. More recent studies in encoder-based personalization [26, 63, 65, 79] propose new training paradigms to condition a Diffusion Model on single or multiple input images. Although these works can enable faster inference, they necessitate extensive pre-training and typically limit application to particular domains. New forms of conditioning, objectives or adaptors are proposed for different purposes such as stylization [23, 29, 51], improved identity preservation [13, 43], composability [37, 56, 82, 85], or generation fidelity [50]. Unlike these works, we focus on addressing an inherently different task of improving the editability of personalized Diffusion Models.

**Image Editing.**  The conditioning mechanisms [14, 24] and inversion techniques [49, 68] have enabled DPMs to achieve image editing. Earlier works [32, 34, 40] relied on heavy optimization and often failed in local editing, while other works [4, 52] achieved this using user-provided mask guidance. Prompt-to-Prompt [21] achieves both local and global editing using cross-attention map injection without requiring extra guidance. Similarly, PnP [72] proposes employing UNet attention features to achieve image-to-image translation. Masactrl [6] extends Prompt-to-Prompt [21] with mutual self-attention to manipulate subject pose or view. Despite the success of these works, most of them are designed only for text-driven editing, leaving image editing with personalized concepts still under-explored. Unlike preliminary attempts [10, 17] that rely on specific baselines, we consider editing with visual concepts acquired by an arbitrary existing personalization method.

**Score Distillation.**  Pooled et al. [57] proposes a novel Score Distillation Sampling (SDS) method that can synthesize 3D assets without requiring 3D training data. For improving sample fidelity and mode coverage of SDS, VSD [74] utilizes LoRA adapters to model a Wasserstein gradient flow, NFSD [31] employs negative prompts, CSD [78] introduces Classifier Guidance. SDS has been improved and extended for different downstream purposes, such as conditional generation of different modality assets [28, 36], text-driven visual editing [22, 67] and text-aligned generation [3].

## 3 Preliminaries

**Diffusion Probabilistic Models (DPMs).** With an input image latent state $\mathbf{x}_0 \in \mathcal{X}$, a corresponding text prompt $\mathbf{y} \in \mathcal{Y}$, and a diffusion process defined as $q\left(\mathbf{x}_t \mid \mathbf{x}_0\right) := \mathcal{N}\left(\mathbf{x}_t; \sqrt{\alpha_t}\mathbf{x}_0, \left(1 - \alpha_t\right)\mathbf{I}\right)$ where $\alpha_t$ represents the forward process variance at time $t$ and $\mathbf{x}_t$ is the noised latent state of the input $\mathbf{x}_0$, a diffusion probabilistic model parameterized by $\phi$ can be trained using the denoising objective:

$$\mathcal{L}_{\mathrm{DPM}}(\mathbf{x}_0, \mathbf{y}; \phi) = \mathbb{E}_{\mathbf{x}_t \sim q(\mathbf{x}_t|\mathbf{x}_0), t \sim \mathcal{U}(1,T), \epsilon \sim \mathcal{N}(\mathbf{0},\mathbf{I})}\left[\|\epsilon_\phi\left(\mathbf{x}_t, \mathbf{y}, t\right) - \epsilon\|_2^2\right]. \tag{1}$$

**T2I Personalization with DPMs.** Given a diffusion model $\epsilon_{\phi_0}$ pre-trained for Text-to-Image (T2I) generation using Eq. 1, and a small set of image latent states and prompt pairs $\mathcal{D}_{\mathcal{X}\mathcal{Y}}^{\mathrm{ref}} = \left\{(\tilde{\mathbf{x}}^{\mathrm{ref}}, \tilde{\mathbf{y}}^{\mathrm{ref}})_n\right\}_{n=1}^N$, which encapsulate the personalized concepts the user wishes the diffusion model to capture.

A general approach of T2I personalization methods [16, 62] is to fine-tune a subset of the source model parameters on $\mathcal{D}_{\mathcal{X}\mathcal{Y}}^{\mathrm{ref}}$ by optimizing the denoising objective as:

$$\phi = \arg\min_{\hat{\phi}} \mathbb{E}_{(\tilde{\mathbf{x}}^{\mathrm{ref}}, \tilde{\mathbf{y}}^{\mathrm{ref}}) \sim \mathcal{D}_{\mathcal{X}\mathcal{Y}}^{\mathrm{ref}}} \mathcal{L}_{\mathrm{DPM}}(\tilde{\mathbf{x}}^{\mathrm{ref}}, \tilde{\mathbf{y}}^{\mathrm{ref}}; \hat{\phi}) \tag{2}$$

where $\hat{\phi}$ is initialized with the pre-trained weights $\phi_0$. The text prompt $\tilde{\mathbf{y}}^{\mathrm{ref}}$ takes the form of "a photo of a $[S]$", where the placeholder $[S]$ corresponds to a new word embedding representing the specific subject. At inference time, the fine-tuned model $\epsilon_\phi$ can be used to generate creative images of the specific subject in novel contexts, such as "a photo of a $[S]$ sitting next to a mirror".

**Score Distillation.** Score Distillation is a mechanism for sampling from a source diffusion model under predefined constraints, achieved by performing probability density distillation from the source diffusion model $\epsilon_\phi$ to a parameterized differentiable function. In the context of image domains, we represent this differentiable function as $x(\theta)$, which renders the image with parameter $\theta$. SDS [57] is the pioneering approach in score distillation, which optimizes $x(\theta)$ through a mode-seeking process. Specifically, given a target prompt $\hat{\mathbf{y}}$ that describes the desired edit, the gradient of the distillation objective w.r.t. parameter $\theta$ is computed as follows:

$$\nabla_\theta \mathcal{L}_{\mathrm{SDS}}(\phi, x(\theta), \hat{\mathbf{y}}) = \mathbb{E}_{t \sim \mathcal{U}(1,T), \epsilon \sim \mathcal{N}(\mathbf{0},\mathbf{I})}\left[\omega(t)\left(\epsilon_\phi\left(x_t(\theta), \hat{\mathbf{y}}, t\right) - \epsilon\right) \frac{\partial x(\theta)}{\partial \theta}\right] \tag{3}$$

where $x_t(\theta)$ is a noised latent state of $x(\theta)$ at time step $t$, and $\omega(t)$ is a constant determined by the forward process variance. SDS is known to be a mode-seeking process and suffers from issues such as low diversity, over-saturation, and over-smoothness [74], which lead to sub-optimal performance on downstream tasks. Thus, various SDS variants have been proposed to enhance performance.

DDS [22] extends SDS to handle text-driven image editing. Specifically, given a source image latent state $\mathbf{x}_0^{\mathrm{src}}$, a source prompt $\mathbf{y}^{\mathrm{src}}$ that is aligned with $\mathbf{x}_0^{\mathrm{src}}$, and a target prompt $\hat{\mathbf{y}}$ that describes the desired edit, the parameter $\theta$ is initialized using the source image such that $x(\theta) = \mathbf{x}_0^{\mathrm{src}}$. For a certain differentiable function $x(\cdot)$, $\theta$ can then be updated using the following delta score direction:

$$\nabla_\theta \mathcal{L}_{\mathrm{DDS}}(\phi, x(\theta), \hat{\mathbf{y}}) = \mathbb{E}_{t \sim \mathcal{U}(1,T), \epsilon \sim \mathcal{N}(\mathbf{0},\mathbf{I})}\left[\omega(t)\left(\epsilon_\phi\left(x_t(\theta), \hat{\mathbf{y}}, t\right) - \epsilon_\phi\left(\mathbf{x}_t^{\mathrm{src}}, \mathbf{y}^{\mathrm{src}}, t\right)\right) \frac{\partial x(\theta)}{\partial \theta}\right]. \tag{4}$$

To simplify the notations, we incorporate all constants associated with score computations into $\omega(t)$ for the remainder of the paper.

**From text-driven editing to personalized editing** Existing text-driven editing works like DDS generate high-quality output for image editing with open-world vocabulary; however, their capability of personalized concepts editing remains underexplored. In this work, we focus on bridging such a gap. Given a diffusion model $\epsilon_\phi$ personalized via Eq. 2 and the latent state of a source image $\mathbf{x}_0^{\mathrm{src}}$ with an aligned source prompt $\mathbf{y}^{\mathrm{src}}$ (e.g., "a photo of a cat sitting next to a mirror"), our objective is to edit $\mathbf{x}_0^{\mathrm{src}}$ using the personalized concept $[s]$ captured by $\epsilon_\phi$. For example, an edit might be "a photo of a (cat $\rightarrow [s]$) sitting next to a mirror," where $[s]$ represents your childhood pet cat. We define the target prompt as $\mathbf{y}^{\mathrm{ref}}$. The desiderata for personalized editing are concluded as follows

- *the overall structure of edited image should align with the source image,*
- *the edited part should capture the appearance and content of the personal subject,*
- *the instruction-irrelevant part should be preserved as much as possible.*

## 3.1 Preliminary Editing with Existing Personalization Baseline

We explore a preliminary solution to personalized editing by employing SDS in Eq. 3 and DDS in Eq. 4 based on the personalized DPM $\epsilon_\phi$, where the differentiable function is initialized by the source image latent state, i.e., $x(\theta) = \mathbf{x}_0^{\mathrm{src}}$, and can be updated with $\nabla_\theta \mathcal{L}_{\mathrm{DDS}}(\phi, x(\theta), \mathbf{y}^{\mathrm{ref}})$. However, this straightforward solution yields unsatisfactory performance as shown in Fig. 5 (b)-(c). Specifically, the editing result with SDS suffers from the same low fidelity issues as mentioned by prior work [22, 74]. With DDS, although the overall layout preservation is improved, there still reveals a lack of editability. Additionally, we observe that DDS leads to a severe bias of certain attributes on the source class, leading to unsatisfactory editing results.

**Source Score Bias Correction.** We suspect that such bias is caused by the distribution shift of $\epsilon_\phi$ during personalization. As shown in Fig. 2, a DreamBooth trained for the concept "plushie_tortoise" shows significant bias towards the yellow color in the corresponding source class "tortoise", which leads to the incorrect color of the edited image as shown in Fig. 5 (c). To address this, we use the pre-trained diffusion model $\epsilon_{\phi_0}$ to conduct the source score prediction in Eq. 4, yielding the modified delta score named as DDS-S, i.e.,

$$\nabla_\theta \mathcal{L}_{\mathrm{DDS\text{-}S}} = \mathbb{E}_{t,\epsilon}\left[\omega(t)\left(\epsilon_\phi\left(x_t(\theta), \mathbf{y}^{\mathrm{ref}}, t\right) - \epsilon_{\phi_0}\left(\mathbf{x}_t, \mathbf{y}^{\mathrm{src}}, t\right)\right)\frac{\partial x(\theta)}{\partial \theta}\right]. \tag{5}$$

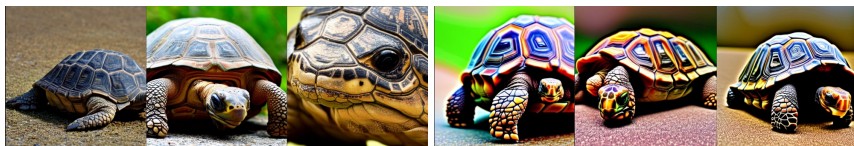

(a) Generations of 'tortoise' on source model. (b) Generations of 'tortoise' on DreamBooth.

Figure 2: Source class bias of DreamBooth trained for "plushie_tortoise".

As shown in Fig. 5 (d), the modified delta score can effectively alleviate the source score bias issue.

# 4 The DreamSteerer Method

## 4.1 Editability Driven Score Distillation (EDSD)

Direct incorporation between an existing personalized model and DDS-S in Eq. 5 relieves the bias caused by inaccurate source score. However, as shown in Fig.5 (a) and (d), a deficiency in editability still persists, evident from the structural misalignment between source and edited images. We hypothesize that this issue stems from the misalignment between score estimations of the pre-trained and personalized models, underscoring the necessity for adjustments to the personalized model. An intriguing direction is to adjust the personalized model such that the objective of the DDS-S is further optimized. For this purpose, we first recover the loss from the gradient form of the DDS-S in Eq. 5 as $\mathcal{L}_{\mathrm{DDS-S}} = \mathbb{E}_{t,\epsilon}[\|\epsilon_\phi\left(x_t(\theta), \mathbf{y}^{\mathrm{ref}}, t\right) - \epsilon_{\phi_0}\left(\mathbf{x}_t^{\mathrm{src}}, \mathbf{y}^{\mathrm{src}}, t\right)\|_2^2]$. For the purpose of editability enhancement, instead of performing score distillation w.r.t. the rendering parameter $\theta$, we perform score distillation w.r.t. the personalized model parameters $\phi$. Specifically, we first introduce a perturbation on the source latent state using the personalized DPM $\epsilon_\phi$ as follows:

$$\hat{x}_t(\phi) := \mathbf{x}_t^{\mathrm{src}} - \sqrt{1-\alpha_t}\underbrace{\left(\epsilon_\phi\left(\mathbf{x}_t^{\mathrm{src}}, y^{\mathrm{ref}}, t\right) - \epsilon\right)}_{\text{Single-step denoising direction}}, \tag{6}$$

which is equivalent to the noised latent state of Tweedie's estimation [11, 15] of the personalized model $\epsilon_\phi$ w.r.t. $\mathbf{x}_t^{\mathrm{src}}$ (Sec. B).

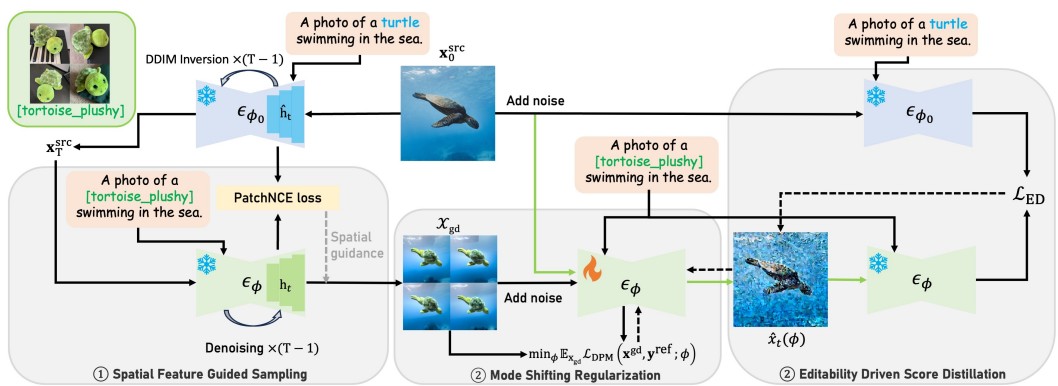

Figure 3: Overall framework of DreamSteerer (the gradient flows are illustrated with dashed lines).

Then, we define the editability-driven loss as $\mathcal{L}_{\text{ED}} = \mathbb{E}_{t,\epsilon}[\|\epsilon_\phi\left(\hat{x}_t(\phi), \mathbf{y}^{\text{ref}}, t\right) - \epsilon_{\phi_0}\left(\mathbf{x}_t^{\text{src}}, \mathbf{y}^{\text{src}}, t\right)\|_2^2]$, which is similar to the $\mathcal{L}_{\text{DDS}-\text{S}}$ but with different parameters to optimize. To reduce the computational cost, with a slight abuse of notation, we use $\nabla_\phi L_{\text{ED}} \approx \frac{\partial \mathcal{L}_{\text{ED}}}{\partial \hat{x}} \frac{\partial \hat{x}}{\partial \phi}$

$$\nabla_\phi \mathcal{L}_{\text{ED}} = \mathbb{E}_{t,\epsilon}\left[\omega(t)\left(\epsilon_\phi\left(\hat{x}_t(\phi), \mathbf{y}^{\text{ref}}, t\right) - \epsilon_{\phi_0}\left(\mathbf{x}_t^{\text{src}}, \mathbf{y}^{\text{src}}, t\right)\right) \frac{\partial \epsilon_\phi\left(\hat{x}_t(\phi), \mathbf{y}^{\text{ref}}, t\right)}{\partial \hat{x}_t(\phi)} \frac{\partial \hat{x}(\phi)}{\partial \phi}\right], \quad (7)$$

where all constants are absorbed in $\omega(t)$. Intuitively, the perturbation introduced by the single denoising step in Eq. 6 can be interpreted as a small editing step on $\mathbf{x}_t^{\text{src}}$ by $\epsilon_\phi$. Therefore, optimizing $\mathcal{L}_{\text{ED}}$ can be understood as distilling information from the source model $\epsilon_{\phi_0}$ into the personalized diffusion model $\epsilon_\phi$ through $\hat{x}_t(\phi)$. Specifically, without bias to the reference dataset, the source model $\epsilon_{\phi_0}$ has better editability on the source image $\mathbf{x}_t^{\text{src}}$ than the personalized model $\epsilon_\phi$, thus its score estimation, i.e., the noising prediction, tends to be more accurate. As $\epsilon_\phi$ is trained on $\mathcal{D}_{\mathcal{XY}}^{\text{ref}}$ via personalization, in order for the personalized model $\epsilon_\phi$ to achieve a similar score estimation, the "edited" $\hat{x}_t(\phi)$ would be prone to have characteristics similar to the personal subject without losing the overall fidelity, through which enhancement in editability can be achieved. In Eq. 7, with parameter sharing, $\epsilon_\phi$ naturally serves as the score estimator for $\hat{x}_t(\phi)$, without requiring additional training. The detailed gradient flow is shown in Fig. 3.

## 4.2 Mode Shifting regularization with spatial feature guided sampling.

**The mode trapping issue of EDSD.** Directly optimizing the personalized model parameter $\phi$ via Eq. 7 results in a peculiar characteristic observed in the edited and generated images, specifically the generated and edited results show patterns that fall between the reference images $\mathcal{D}_{\mathcal{XY}}^{\text{ref}}$ and the source image $\mathbf{x}_0^{\text{src}}$. As shown in Fig. 4 (f), when using a silver cat image as source and a brown cat as reference subject, the generated images reveal a hybrid appearance compared to the source model generations in Fig. 4 (e), showcasing features of both silver and brown cats. This observation suggests that EDSD has induced $\epsilon_\phi$ to collapse to a trivial trapping point between the modes of the source image and the reference images.

**Spatial feature guided sampling.** To avoid such a mode trapping issue and maintain the concept of the personal subject, we regularize EDSD by jointly training the model on a set of personal subject images. Instead of accessing the reference dataset, we use images generated by the personalized model $\epsilon_\phi(\cdot \mid \mathbf{y}^{\text{ref}})$. We find that guiding the generated samples to have a structural layout akin to the source image $\mathbf{x}_0^{\text{src}}$ not only serves as an effective regularization but also shifts the model distribution $p_\phi(\cdot \mid \mathbf{y}^{\text{ref}})$ towards a more editable region than the original mode centered around reference images.

Recent works [6, 72] show that the Self-Attention (SA) features of the T2I DPMs are embedded with detailed spatial information, thus having a strong sense of spatial layout that allows building inter-image semantic correspondence using these features [1, 80]. Motivated by these findings, we propose a spatial feature guided sampling strategy using these features from the source image.

As shown in Fig. 3, we begin by performing the DDIM inversion [68] on the source image using the pre-trained DPM $\epsilon_{\phi_0}$. At each time step $t$, for the $\ell$-th SA layer from $\epsilon_{\phi_0}$, an intermediate feature

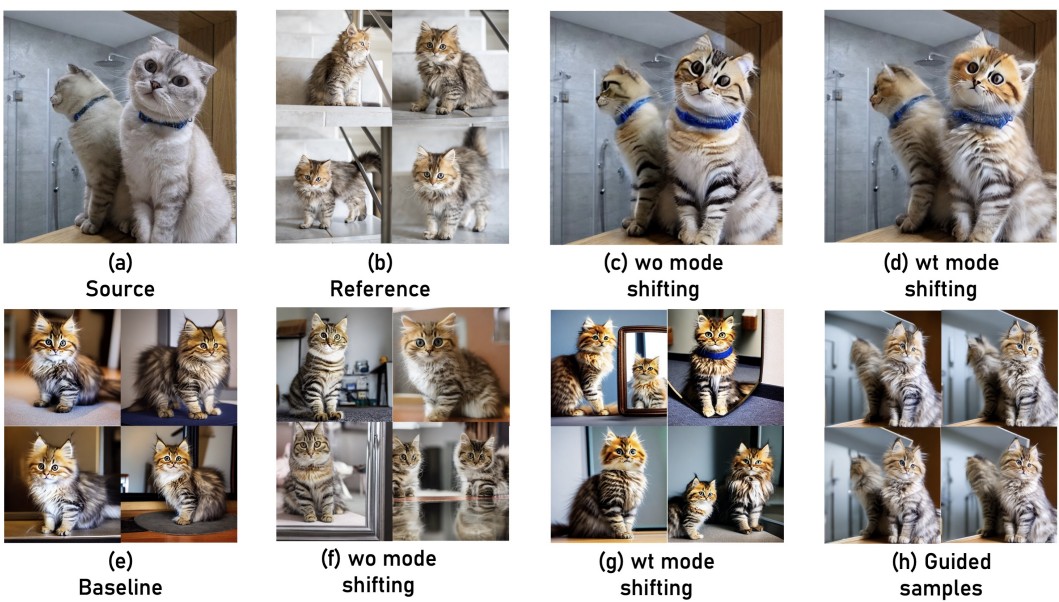

Figure 4: The effect of different regularization strategies on the editing and generation results of a DreamBooth baseline. The source prompt is "a photo of a cat sitting next to a mirror".

vector $\psi_\ell(\mathbf{x}_t)$ is extracted, which is further projected linearly into queries $Q_t^\ell = f_Q^\ell(\psi_\ell(\mathbf{x}_t))$, keys $K_t^\ell = f_K^\ell(\psi_\ell(\mathbf{x}_t))$ and values $V_t^\ell = f_V^\ell(\psi_\ell(\mathbf{x}_t))$. The final output SA feature is computed via $\hat{\mathbf{h}}_t^\ell = \text{Softmax}\left(Q_t^\ell \left(K_t^\ell\right)^T / \sqrt{C^\ell}\right) V^\ell$, where $C^\ell$ is the channel size of the keys and queries. The output spatial features $\hat{\mathbf{h}}_t^\ell \in \mathbb{R}^{S^\ell \times C^\ell}$ are encoded with localized semantic information, where $S^\ell$ is the number of spatial locations in the $\ell$-th layer. These features are cached and serve as guidance for sampling with the personalized model $\epsilon_\phi(\cdot \mid \mathbf{y}^{\text{ref}})$. Specifically, after running DDIM inversion with the source model $\epsilon_{\phi_0}(\cdot \mid \mathbf{y}^{\text{src}})$, we obtain the inverted initial latent state of the source image $\mathbf{x}_T^{\text{src}}$, starting from which the reverse diffusion process is conducted using the personalized model $\epsilon(\cdot \mid \mathbf{y}^{\text{ref}})$. At each time step $t$, the same set of spatial features $\{\mathbf{h}_t^\ell\}$ are extracted, and we aim to condition the model prediction on the source image $\mathbf{x}_0^{\text{src}}$ by matching the pairs $\mathbf{h}_t^\ell$ and $\hat{\mathbf{h}}_t^\ell$ at corresponding spatial locations. For this purpose, following previous work [53, 84], we compute the patchwise contrastive loss between the spatial features as follows

$$\mathcal{L}_{\text{PatchNCE}}(\mathbf{h}_t, \hat{\mathbf{h}}_t) = \sum_{l=1}^{L} \sum_{s=1}^{S_\ell} \mathcal{L}_{\text{NCE}}\left(\mathbf{h}_t^{\ell,s}, \hat{\mathbf{h}}_t^{\ell,s}, \hat{\mathbf{h}}_t^{\ell,S_\ell \setminus s}\right), \tag{8}$$

where we treat the spatial feature vectors from the same spatial location as positive pairs, those from different spatial locations are treated as negative pairs and $\mathcal{L}_{\text{NCE}}(\mathbf{h}, \mathbf{h}^+, \mathbf{h}^-) = -\log\left[\frac{\exp(\mathbf{h} \cdot \mathbf{h}^+/\tau)}{\exp(\mathbf{h} \cdot \mathbf{h}^+/\tau) + \sum_{n=1}^{N} \exp(\mathbf{h} \cdot \mathbf{h}_n^-/\tau)}\right]$ with some coefficient $\tau > 0$. With Eq. 8, for the generation with the personalized model at time step $t$, we model the likelihood of a noisy latent state $\mathbf{x}_t$ matching the structural layout of the source image at time step t as $\nabla_{\mathbf{x}_t} \log p_\phi(\hat{\mathbf{h}}_t \mid \mathbf{x}_t) \approx \nabla_{\mathbf{x}_t} \mathcal{L}_{\text{PatchNCE}}(\mathbf{h}_t, \hat{\mathbf{h}}_t)$. Following the Classifier Guidance [14], we modify the noise prediction of Classifier-Free Guidance as follows

$$\begin{aligned}
\epsilon_\phi^{\text{gd}}\left(\mathbf{x}_t, \mathbf{y}^{\text{ref}}, \hat{\mathbf{h}}_t\right) &= \epsilon_\phi^{\text{cfg}}\left(\mathbf{x}_t, \mathbf{y}^{\text{ref}}\right) - \sigma_t \lambda \nabla_{\mathbf{x}_t} \mathcal{L}_{\text{PatchNCE}}\left(\mathbf{h}_t, \hat{\mathbf{h}}_t\right) \\
&\approx -\sigma_t \nabla_{\mathbf{x}_t} \left[\log p_\phi\left(\mathbf{x}_t \mid \mathbf{y}^{\text{ref}}\right) + \lambda \log p_\phi\left(\hat{\mathbf{h}}_t \mid \mathbf{x}_t\right)\right],
\end{aligned} \tag{9}$$

where $\lambda$ is a weight that balances two scores. Using such a spatial feature guided sampling strategy, we can sample a set of images $\mathcal{X}^{\text{gd}}$. As shown in Fig. 4 (h), these images are prone to have a similar structure to the source image without losing the appearance of the personal concept [s].

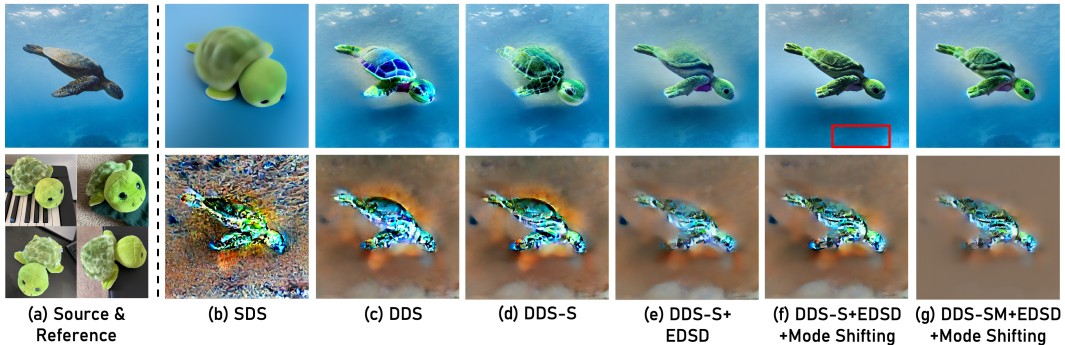

| (a) Source & Reference | (b) SDS | (c) DDS | (d) DDS-S | (e) DDS-S+ EDSD | (f) DDS-S+EDSD +Mode Shifting | (g) DDS-SM+EDSD +Mode Shifting |

Figure 5: Illustration on the effect of the proposed components on editing with a DreamBooth baseline (1st row shows the editing results; 2nd row shows the editing directions, where brown means zero).

Table 1: Comparison with different baselines (DreamSteerer uses the same model as baseline).

| Method | CLIP-I (↑) | | LPIPS (↓) | | SSIM (↑) | MS-SSIM (↑) | IQA (↑) | | |
|---|---|---|---|---|---|---|---|---|---|
| | CLIP B/32 | CLIP L/14 | Alex | VGG | | | Topiq | Musiq | LIQE |
| Textual Inversion [16] | 0.785 | 0.746 | 0.133 | 0.181 | 0.823 | 0.880 | .577 | 68.4 | 4.23 |
| **DreamSteerer** | **0.788** (+.4%) | **0.753** (+.9%) | **0.115** (-13.4%) | **0.167** (-8.2%) | **0.833** (+1.3%) | **0.898** (+2.0%) | **.593** (+2.8%) | **70.1** (+2.5%) | **4.39** (+3.8%) |
| Custom Diffusion [39] | 0.794 | 0.751 | 0.204 | 0.247 | 0.755 | 0.785 | .604 | 70.4 | 4.35 |
| **DreamSteerer** | **0.796** (+.2%) | **0.761** (+1.3%) | **0.183** (-10.2%) | **0.229** (-7.3%) | **0.768** (+1.8%) | **0.815** (+3.7%) | **.608** (+.6%) | **71.9** (+2.1%) | **4.44** (+2.1%) |
| Custom Diffusion [39] | 0.781 | 0.743 | 0.199 | 0.239 | 0.767 | 0.808 | .591 | 69.8 | 4.24 |
| **DreamSteerer** | **0.784** (+.29%) | **0.746** (+.7%) | **0.182** (-8.9%) | **0.227** (-5.1%) | **0.779** (+1.5%) | **0.833** (+3.1%) | **.612** (+3.6%) | **71.5** (+2.4%) | **4.41** (+4.%) |

**Mode shifting regularization.** With the set of guided samples $\mathcal{X}_{\mathrm{gd}}$, a mode shifting regularization term is jointly optimized with EDSD as

$$\phi \leftarrow \phi - \eta\left(\nabla_\phi \mathcal{L}_{\mathrm{ED}} + \nabla_\phi \mathcal{L}_{\mathrm{MS}}\right), \ \mathcal{L}_{\mathrm{MS}} := \mathbb{E}_{\mathbf{x}^{\mathrm{gd}} \sim \mathcal{X}^{\mathrm{gd}}} \mathcal{L}_{\mathrm{DPM}}\left(\mathbf{x}^{\mathrm{gd}}, \mathbf{y}^{\mathrm{ref}}; \phi\right), \quad (10)$$

where $\eta$ is the learning rate. As shown in Fig. 4 (g), the mode trapping issue in Fig. 4 (f) can be avoided with the mode shifting regularization in Eq. 10, where the generated images maintain appearance fidelity comparable to source model generations in Fig. 4 (e); meanwhile, the generations exhibit patterns akin to the source image. e.g., features like the blue collar, the presence of "two cats" and subject pose closely resemble those in the source image in Fig. 4 (a), indicating that the mode of $p_\phi(\cdot)$ has been effectively steered to enhance editability for the source image. Furthermore, this enhancement is evidenced by the noticeable improvement in appearance fidelity of edited images from Fig. 4 (c) to Fig. 4 (d). See Fig. 3 for an overall framework.

### 4.3 Automatic Subject Masking

Given the final steered personalized model $\epsilon_{\tilde{\phi}}$ after optimization through Eq. 10, although Eq. 5 leads to pleasant target concept alignment, we observe that the subject irrelevant part may not be properly maintained due to the structural layout difference between source and target subjects as shown in Fig. 5. Inspired by recent work [21, 70] showing that the Cross-Attention (CA) maps concentrate on the relevant regions of the corresponding prompt token, we automatically extract subject masks $M(\mathbf{x}_0^{\mathrm{src}})$ (refer to Sec. J for details) and we define such masked delta score as DDS-SM:

$$\nabla_\theta \mathcal{L}_{\mathrm{DDS-SM}} = \mathbb{E}_{t,\epsilon}\left[\omega(t) M(\mathbf{x}_0^{\mathrm{src}}) \odot \left(\epsilon_{\tilde{\phi}}\left(x_t(\theta), \mathbf{y}^{\mathrm{ref}}, t\right) - \epsilon_{\phi_0}\left(\mathbf{x}_t, \mathbf{y}^{\mathrm{src}}, t\right)\right) \frac{\partial x(\theta)}{\partial \theta}\right], \quad (11)$$

which better focuses on editing the subject-relevant regions as shown in Fig. 5 (g).

## 5 Experiments

**Evaluation metrics.** In accordance with the desired editing properties discussed in Sec. 3, we evaluate the effectiveness of DreamSteerer from three perspectives: 1) semantic similarity with the reference images using CLIP image similarity with CLIP ViT-B/32 and CLIP ViT-L/14 [58], 2) perceptual similarity with the source image using LPIPS with AlexNet [38] and VGG [66], 3) structural similarity with the source image using SSIM [76] and MS-SSIM [75]. Additionally, to validate the editing fidelity of our proposed method, we report the No-Reference Image Quality Assessment (IQA) metrics Topiq [8], Musiq [33] and LIQE [81].

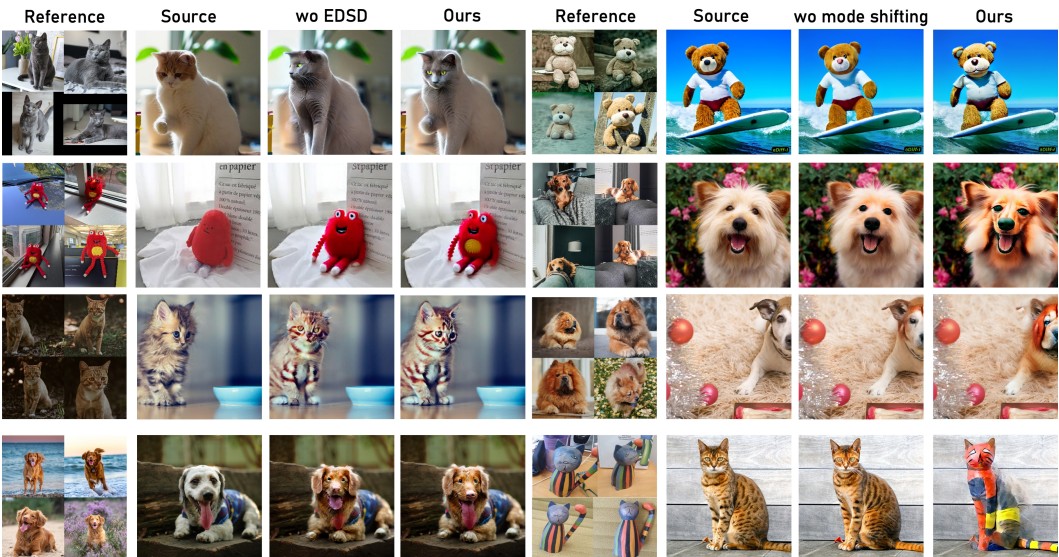

Figure 6: Ablation study on EDSD and Mode Shifting Regularization.

Table 2: Ablation study on EDSD and Mode Shifting, the best and second best results are highlighted.

| Baseline | EDSD | Mode Shifting | CLIP-I (↑) | | LPIPS (↓) | | SSIM (↑) | MS-SSIM (↑) |
|---|---|---|---|---|---|---|---|---|
| | | | CLIP B/32 | CLIP L/14 | Alex | VGG | | |
| Textual Inversion [16] | ✓ | ✓ | 0.788 | 0.753 | 0.115 | 0.167 | 0.833 | 0.898 |
| | ✗ | ✓ | 0.788 | 0.749 | 0.132 | 0.182 | 0.821 | 0.880 |
| | ✓ | ✗ | 0.779 | 0.746 | 0.090 | 0.144 | 0.850 | 0.922 |
| DreamBooth [62] | ✓ | ✓ | 0.796 | 0.761 | 0.182 | 0.229 | 0.768 | 0.815 |
| | ✗ | ✓ | 0.794 | 0.755 | 0.200 | 0.246 | 0.753 | 0.792 |
| | ✓ | ✗ | 0.782 | 0.745 | 0.134 | 0.186 | 0.805 | 0.869 |
| Custom Diffusion [39] | ✓ | ✓ | 0.784 | 0.746 | 0.182 | 0.227 | 0.779 | 0.833 |
| | ✗ | ✓ | 0.779 | 0.737 | 0.217 | 0.262 | 0.748 | 0.795 |
| | ✓ | ✗ | 0.760 | 0.730 | 0.100 | 0.152 | 0.841 | 0.917 |

**Implementation details.** We evaluate our plug-in method on three personalization baselines: Textual Inversion [16], DreamBooth [62] and Custom Diffusion [39], which include the 3 mainstream types of models outlined in Sec. 2. We use the pre-trained checkpoints provided by Dream-Matcher [50], with 16 concepts for each baseline encompassing living, non-living, in-door, and outdoor subjects. For each baseline, 70 random real-world images are used, focusing on the challenging editing scenarios as shown in Fig 1. For a fair comparison, all experiments use DDS-SM (Eq. 11) as the editing method.

**Comparison with baseline methods.** Table. 1 compares DreamSteerer against baselines [16, 39, 62]. Our work shows clear improvement in all 3 types of metrics, with substantial gains in the perceptual and structural alignment with the source images. Even for challenging editing scenarios (see Fig. 1) such as mirror reflections and significant structural changes, where baseline methods often result in distortions and unfaithful structure maintenance, our method effectively calibrates these issues to achieve high-fidelity results. Refer to Supp. F for more editing results. We find that the automatic metrics do not fully reflect the superior performance of our method compared to the baselines, particularly in terms of the quality of edited images. Therefore, we conduct a user preference study using the same criteria in Supp. L; our work is preferred by the users by a significant margin against the baselines.

**Ablation study.** We ablate EDSD and Mode Shifting Regularization to demonstrate their effectiveness in our framework. As shown in Table. 2 and Fig. 6, without EDSD, the CLIP-I scores remain comparable to those of the full model. However, the performance in source image alignment deteriorates significantly, as indicated by the lower LPIPS and SSIM scores. Without Mode Shifting

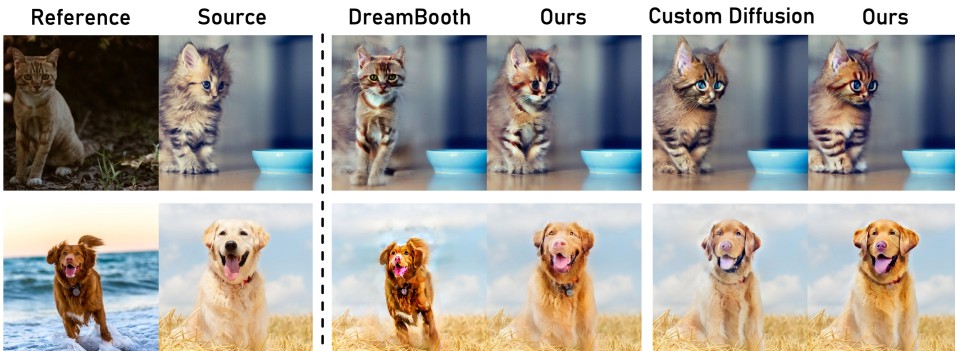

Figure 7: Comparison of one-shot performance.

Table 3: Comparison with baselines under one-shot scenario.

| Method | CLIP-I (↑) | | LPIPS (↓) | | SSIM (↑) | MS-SSIM (↑) |
|---|---|---|---|---|---|---|
| | CLIP B/32 | CLIP L/14 | Alex | VGG | | |
| DreamBooth [62] | 0.790 | 0.717 | 0.222 | 0.264 | 0.742 | 0.777 |
| **DreamSteerer** | **0.801** (+1.4%) | **0.748** (+4.3%) | **0.160** (-27.9%) | **0.212** (-16.7.%) | **0.781** (+5.2%) | **0.847** (+9.0%) |
| Custom Diffusion [39] | 0.796 | 0.740 | 0.155 | 0.210 | 0.782 | 0.856 |
| **DreamSteerer** | **0.799** (+.4%) | **0.753** (+1.8%) | **0.145** (-6.4%) | **0.200** (-4.7%) | **0.796** (+1.7%) | **0.873** (+1.9%) |

regularization, exceptionally high structural and perceptual alignment scores are achieved. However, as depicted in Fig. 6, this often results in a severe loss of target subject appearance information, reflected in consistently poor CLIP-I scores. Overall, combining EDSD and Mode Shifting Regularization achieves the best trade-off between source image alignment and target concept alignment. Fig. 5 provides a component analysis, showing that EDSD effectively improves the editability of the baseline DreamBooth model, resulting in higher structural alignment with the source image in terms of the edited results and the editing directions (refer to Supp. I).

**One-shot performance.** We further evaluate the performance of DreamSteerer against baselines under an extreme data-hungry scenario of one-shot personalization. We observe that Textual Inversion, which does not update the Diffusion Model parameters, cannot provide valid editing results under this setting. Therefore, DreamBooth and Custom Diffusion are used. As shown in Table 3, DreamSteerer maintains superior performance under these conditions. Notably, DreamBooth, relying on full fine-tuning, exhibits a severe bias towards the pose and structure of the reference image. Despite such strong bias, DreamSteerer improves its performance by a significant margin, as shown in Fig. 7.

**More comparisons with existing works.** See Sec. D for a discussion on the setting-level differences between our work and existing subject swapping works [17, 42]. We use a modified Delta Denoising Score as the base editing model, as this method provides stable editing results with all the personalized models we use. However, DreamSteerer is not restricted to a specific type of editing pipeline. We also evaluate the effectiveness of our method as a plug-in for personalized editing with Prompt-to-Prompt [21]. See Sec. D for a comparison and a discussion on how Prompt-to-Prompt may be incompatible with certain base personalization models due to the limitations of the existing latent state inversion method.

## 6  Conclusion

In this work, we identify that existing T2I personalization models fail to deliver satisfactory image editing results. Therefore, we present DreamSteerer, an efficient plug-in method designed to enhance the editability of images conditioned on the source image. DreamSteerer fine-tunes the personalization parameters by training a novel Editability Driven Score Distillation objective under the constraint of a Mode Shifting regularization term based on spatial feature-guided samples. Through experiments, we show that DreamSteerer can significantly improve the editing fidelity of various existing baselines, particularly in challenging editing cases and data-hungry personalization scenarios. We consider DreamSteerer as a pivotal bridge from text-driven image editing to personalized image editing.

# 7 Acknowledgement

Special thanks to Peter Tu for providing us with his valuable inputs and support that greatly helped improving our work and for motivating and encourage us on further implementations of our work. This research was, in part, funded by the U.S. Government – DARPA ECOLE HR00112390061 and DARPA TIAMAT HR00112490421. The views and conclusions contained in this document are those of the authors and should not be interpreted as representing the official policies, either expressed or implied, of the U.S. Government.

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

# A  More Background on Diffusion Probablistic Models

**Denoising Diffusion Implicit Model (DDIM).**   Given a diffusion probabilistic model parameterized by $\phi$ and a diffusion process defined as $q\left(\mathbf{x}_t \mid \mathbf{x}_0\right) := \mathcal{N}\left(\mathbf{x}_t; \sqrt{\alpha_t}\mathbf{x}_0, \left(1 - \alpha_t\right)\mathbf{I}\right)$ where $\alpha_t$ represents the forward process variance at time $t$ and $\mathbf{x}_t$ is the noised latent state of the input $\mathbf{x}_0$, DDIM [68] defines the following update rule in the reverse diffusion process

$$\mathbf{x}_{t-1} = \sqrt{\alpha_{t-1}} \underbrace{\left( \frac{\mathbf{x}_t - \sqrt{1 - \alpha_t}\epsilon_\phi^{(t)}\left(\mathbf{x}_t\right)}{\sqrt{\alpha_t}} \right)}_{\text{"predicted } \mathbf{x}_0\text{"}} + \underbrace{\sqrt{1 - \alpha_{t-1} - \sigma_t^2}\epsilon_\phi^{(t)}\left(\mathbf{x}_t\right)}_{\text{"direction pointing to } \mathbf{x}_t\text{"}} + \underbrace{\sigma_t\epsilon_t}_{\text{"random noise"}} , \qquad (12)$$

where $\sigma_t$ is a free variable that controls the stochasticity in the reverse process.

**DDIM Inversion.**   By setting $\sigma_t$ to 0, we obtain a deterministic update rule which can be reversed to give a deterministic mapping between $\mathbf{x}_0$ and its latent state $\mathbf{x}_T$. We refer this inverse mapping as DDIM inversion:

$$\frac{\mathbf{x}_{t+1}}{\sqrt{\alpha_{t+1}}} - \frac{\mathbf{x}_t}{\sqrt{\alpha_t}} = \left( \sqrt{\frac{1 - \alpha_{t+1}}{\alpha_{t+1}}} - \sqrt{\frac{1 - \alpha_t}{\alpha_t}} \right) \epsilon_\phi^{(t)}\left(\mathbf{x}_t\right). \qquad (13)$$

**Classifier-free Guidance (CFG).**   Given a diffusion model jointly trained on conditional and unconditional embeddings, at inference time, samples can be generated using CFG [24]. The prediction with the conditional and unconditional estimates are defined as

$$\epsilon_\phi^{\text{cfg}}\left(\mathbf{x}_t, \mathbf{y}, t\right) := \beta\epsilon_\phi\left(\mathbf{x}_t, \mathbf{y}, t\right) + (1 - \beta)\epsilon_\phi\left(\mathbf{x}_t, \varnothing, t\right) , \qquad (14)$$

where $\beta$ is the guidance scale that controls the trade-off between mode coverage as well as sample fidelity and $\varnothing$ is a null token used for unconditional prediction.

# B  Details about single-step denoising direction

We parameterize the single-step perturbed latent state $\hat{x}_t(\phi)$ based on Tweedie's estimation [15], which computes the posterior estimation for the denoised data, i.e., $\hat{\mathbf{x}}_0 := \mathbb{E}[\mathbf{x}_0 \mid \mathbf{x}_t] = \left(\mathbf{x}_t + (1 - \alpha_t)\nabla_{\mathbf{x}_t} \log p(\mathbf{x}_t)\right)/\sqrt{\alpha_t}$, with which we define single-step perturbed latent state $\hat{x}_t(\phi)$ as follows

$$\hat{x}_t(\phi) := \sqrt{\alpha_t}\hat{\mathbf{x}}_0 + \sqrt{1 - \alpha_t}\epsilon = \mathbf{x}_t - \sqrt{1 - \alpha_t} \underbrace{\left(\epsilon_\phi\left(\mathbf{x}_t, y^{\text{ref}}\right) - \epsilon\right)}_{\textit{Single-step denoising direction}} , \qquad (15)$$

where the same noise added to $\mathbf{x}_t$ and $\hat{x}_t(\phi)$.

# C  More Implementation details

**Dataset**   The pre-trained checkpoints provided by ViCo [20] have 16 concepts, which include 6 toys, 5 live animals, 2 types of accessories, 2 types of containers, and 1 building. The corresponding text prompts of source images are generated by BLIP2 [41]. For editing with one-shot personalization, performance evaluations are conducted on a subset of 20 cat and dog images.

**Editability Driven Score Distillation**   The EDSD gradient in Eq. 7 includes a Jacobian term $\frac{\partial \epsilon_\phi\left(\hat{x}_t(\phi), \mathbf{y}^{\text{ref}}\right)}{\partial \hat{x}_t(\phi)}$, which is computationally expensive and unstable for optimization. Previous work [57] shows that this term can cause instability in optimization and omits the computation of it by setting it to $\mathbf{I}$. As shown in Fig. 8 we find that by setting it to $\mathbf{I}$ following previous work, the editing process tends to destroy the structural layout and background of the source image. Meanwhile, the edited image tends to be over-saturated. This indicates that by setting $\frac{\partial \epsilon_\phi\left(\hat{x}_t(\phi), \mathbf{y}^{\text{ref}}\right)}{\partial \hat{x}_t(\phi)} = \mathbf{I}$, the discrepancy between personalized and source model score estimations has been maximized. We find that setting $\frac{\partial \epsilon_\phi\left(\hat{x}_t(\phi), \mathbf{y}^{\text{ref}}\right)}{\partial \hat{x}_t(\phi)} = -\mathbf{I}$ leads to significantly better results with natural adaptation to the layout of the

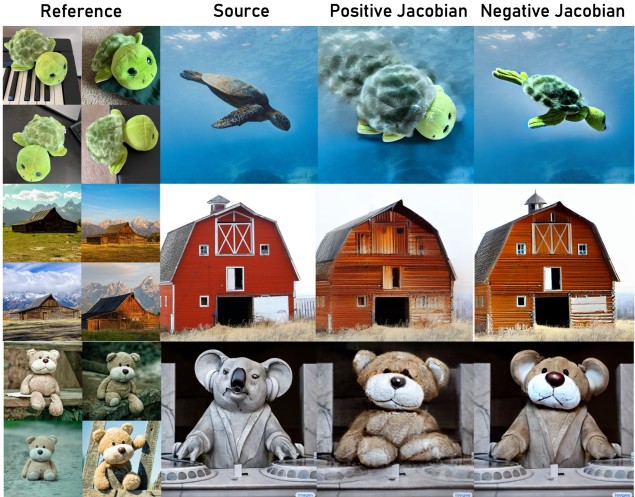

| Reference | Source | Positive Jacobian | Negative Jacobian |

Figure 8: Results with different Jacobian omitting strategy.

source image. We suspect that this is because, in the former work [57], the input noisy latent state is independent from the diffusion model. However, in Eq. 7, there exists an entanglement between negative score estimation in the single-step denoising function (Eq. 6) and the score estimation. We leave a further investigation on this interesting scenario to future study.

**Spatial feature guided sampling**  During spatial feature guided sampling, to avoid potential numerical instability in the sampling process due to the drift in statistics caused by the additional guidance gradient, we rescale the noise prediction via an adaptive normalization [27], i.e., $\tilde{\epsilon}_\phi^{\text{gd}}\left(\mathbf{x}_t, \mathbf{y}^{\text{ref}}, \hat{\mathbf{h}}_t\right) = \text{AdaIN}\left(\epsilon_\phi^{\text{gd}}\left(\mathbf{x}_t, \mathbf{y}^{\text{ref}}, \hat{\mathbf{h}}_t\right), \epsilon_\phi^{\text{cfg}}\left(\mathbf{x}_t, \mathbf{y}^{\text{ref}}\right)\right)$. Following earlier works [83], we introduce spatial feature guidance only for the early denoising steps $t > t_{\text{early}}$. After $t_{\text{early}}$, we change the ODE sampler [68] to an SDE sampler [25] to inject stochasticity for improved sample quality [30]. we employ DDIM with a total step of 50, the CFG is set as 1 for the inversion process, 3.5 for the sampling process, and negative prompts "oversaturated color, ugly, tiling, low quality, noisy" are employed to replace the null text token. We set the early stopping step as $t_{\text{early}} = 30$. The size of $\mathcal{X}_{\text{gd}}$ is set to be the same as the size of the reference images.

**Optimization details**  For fine-tuning with each baseline, we employ the same set of trainable parameters as the original personalization process. We use an AdamW [46] optimizer for all baselines; the learning rates are set as 1e-3, 1e-6, and 5e-5 for Textual Inversion, DreamBooth, and Custom Diffusion respectively. The total optimization step is set to 10 with 10 cumulative gradient steps. Experiments run on a single NVIDIA RTX3090 GPU take a fine-tuning time of around 1 minute for a batch size of 1. For all experiments, we use a latent space diffusion model [61], with pre-trained checkpoints from Stable-Diffusion-v-1-4 [3], which is augmented with safety modules to mitigate NSFW content.

# D   Further comparison with existing works

**Difference with subject swapping**  Text-driven image editing methods generally fall into two categories: rigid editing [5, 21] that emphasize the preservation of the source image, and non-rigid editing [6] that focus on changing the view or pose of a subject in the source image while preserving the background. Our work is motivated by bridging the gap in editability between specific concept and natural language conditioning for rigid editing. Subject swapping methods like DreamEdit [42] and Photoswap [17] diverge from our approach in their main criteria. While these methods prioritize alignment with the source subject's location and pose, they do not necessitate maintaining the original structural details as our method does. Furthermore, these works demand a stricter preservation of

---

[3]Source from https://huggingface.co/CompVis/stable-diffusion-v-1-4-original.

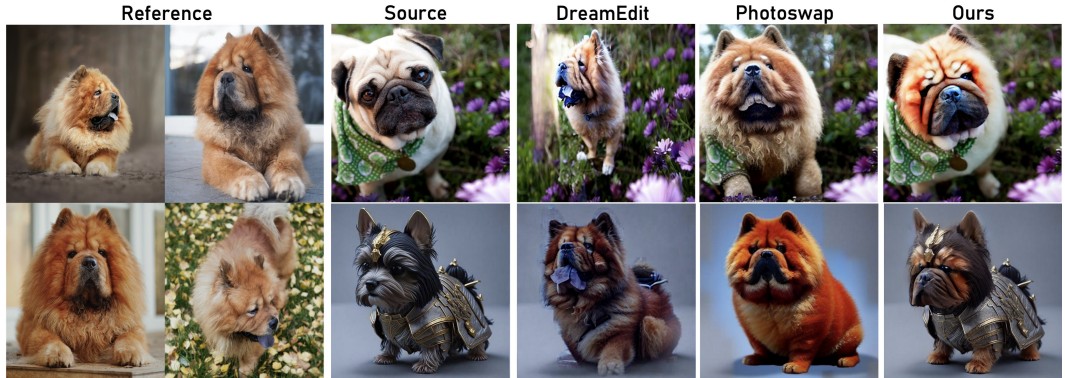

Figure 9: Comparison with subject swapping methods.

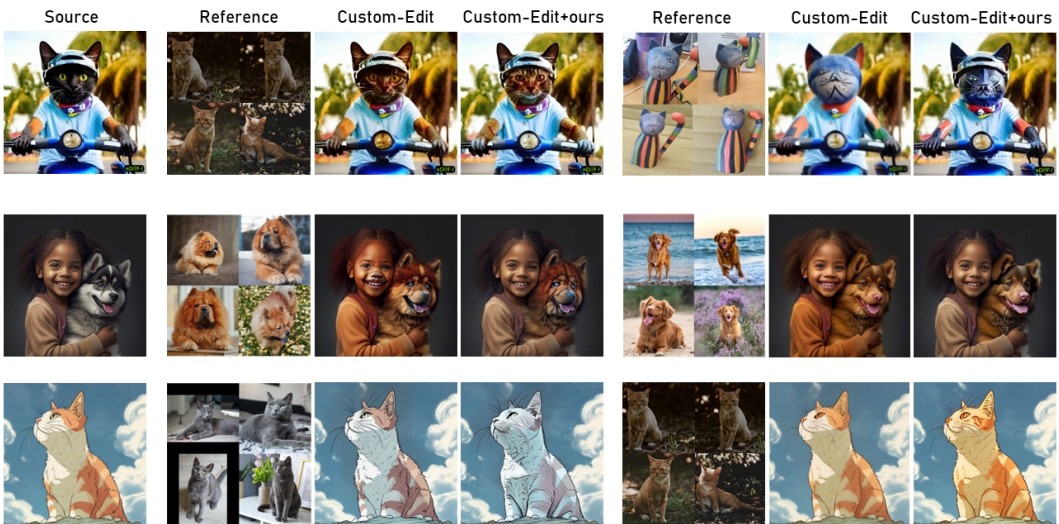

Figure 10: Comparison with Custom-Edit.

subject identity and typically do not require a significant level of concept extrapolation like our work, e.g., from a short cat to a tall cat. In Fig. 9, we provide a comparison with DreamEdit and Photoswap in scenarios involving significant structural gaps. Compared with our work, these works often exhibit severe distortion or fail to maintain structural consistency with the source image.

**Comparison using different image editing method**  We employ a variant of Delta Denoising Score as the base editing model as this method is compatible with all the personalized models we use. However, DreamSteerer is not restricted to a specific type of editing pipeline. We also evaluate the effectiveness of our method as a plug-in for Custom-Edit [10], which directly combines Custom Diffusion with Prompt-to-Prompt [21]. The results are shown in Fig. 10 and Table. 4. Meanwhile, as shown in Fig. 11, combining Prompt-to-Prompt with DreamBooth can introduce significant appearance artifacts in the edits, which is the main reason we did not use it as the base editing method. Prompt-to-Prompt relies on a source latent state inversion process, typically through Null-Text Inversion (NTI). However, parameter updates during personalization can shift the model distribution for the source class, compromising the editability of the inverted latent state chain with NTI. In comparison, the Delta Denoising Score-based edited method employed in our work does not require an inversion process, providing more robust performance across different types of personalization baselines. We believe this phenomenon is worth further investigation and encourage future works to develop new inversion techniques specifically tailored for the personalized models.

Table 4: Comparison with Custom-Edit.

| Method | CLIP-I (↑) | | LPIPS (↓) | | SSIM (↑) | MS-SSIM (↑) | IQA (↑) | | |
| | CLIP B/32 | CLIP L/14 | Alex | VGG | | | Topiq | Musiq | LIQE |
|---|---|---|---|---|---|---|---|---|---|
| Custom Edit [10] | .748 | .727 | .141 | .210 | .793 | .899 | .564 | 67.94 | 3.97 |
| **DreamSteerer** | **.750** | **.729** | **.141** | **.209** | .793 | .899 | **.565** | **67.95** | **3.98** |

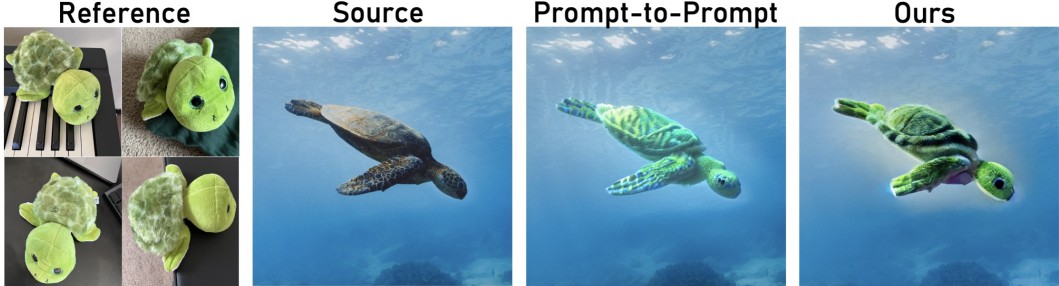

Figure 11: Failure of Prompt-to-Prompt when integrated with DreamBooth.

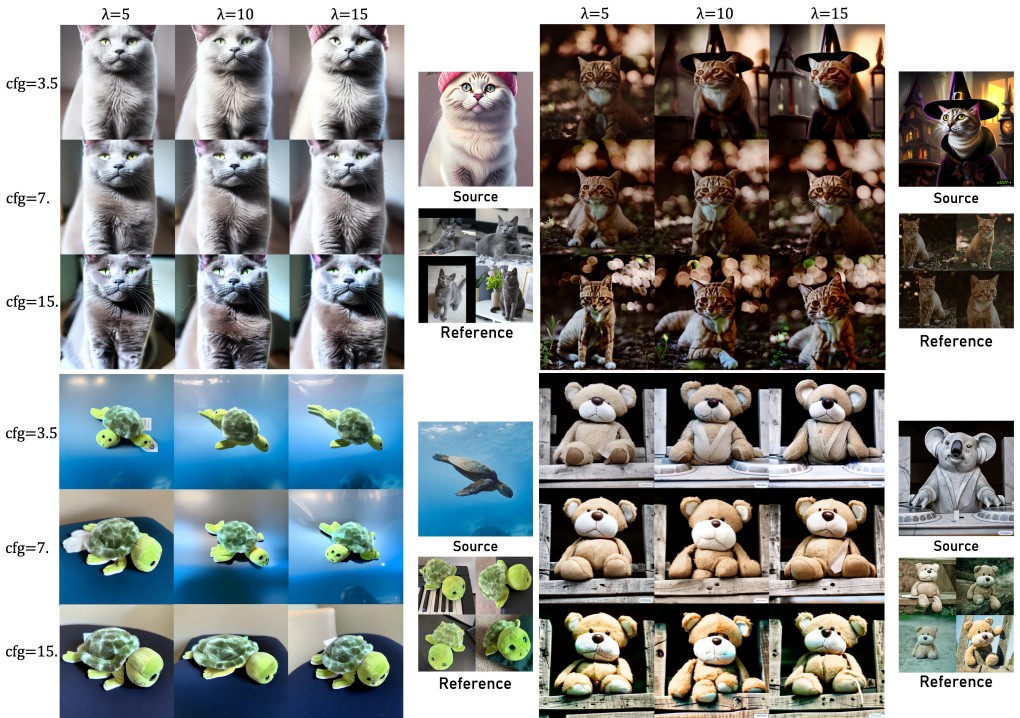

Figure 12: Sensitivity analysis on $\lambda$ and cfg for spatial feature guided sampling

# E   More ablation study

More results of ablation study on EDSD and Most Shifting regularization are provided in Fig. 12

**Sensitivity analysis on spatial feature guided sampling**   We ablate on two hyper-parameters $\lambda$ and cfg introduced in Eq. 9 for spatial feature guided sampling. As shown in Fig. 13, in contrast to the usual selection of large cfg, e.g., 7 or 15. A relatively small cfg of 3.5 leads to better naturalness of the generated samples that avoids obvious subject appearance drift from the reference images. Meanwhile, a large spatial feature guidance scale $\lambda$ produces samples that better preserve the structure of the source image. Thus we set cfg as 3.5 and $\lambda$ as 15.

# F   More results compared with the base personalization methods

More results compared with the baseline models are provided in Fig. 14 15 16.

# G   More results with one-shot personalization

More comparison of one-shot performance with the DreamBooth baseline and Custom Diffusion baseline is provided in Fig. 17

# H   More spatial feature guided sampling results

More spatial feature guided sampling results used in Eq. 9 are provided in Fig. 18, Fig. 19 and Fig. 20.

# I   Explanation on editing direction visualization

In Fig. 5, we visualize the score-based editing directions obtained by taking the editing scores as input to the Stable Diffusion decoder. This is a similar visualization to Katzir et al. [31]. We note that the brown color in the visualization indicates an editing score value of 0. An example of zero-valued latent state decoding is illustrated in Fig. 21.

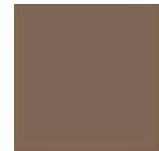

Figure 21: Zero-valued latent decoding.

# J   Automatic subject mask extraction using cross-attention

As shown in Fig. 22, we observe that the attention maps from decoder layers with resolutions 16 and 32 have accurate subject layout. Thus, we use the averaged CA maps from these layers as the subject mask. Similar to the spatial feature extraction process in Sec. 4.2, DDIM inversion is conducted on the source image $\mathbf{x}_0^{\mathrm{src}}$, by which the averaged CA map that corresponds to the source subject token is extracted, i.e.,

$$M(\mathbf{x}_0^{\mathrm{src}}) = \mathbb{E}_{t,\ell} \, \mathrm{Softmax} \left( Q_t^\ell \left( K_t^\ell \right)^T / \sqrt{C^\ell} \right).$$

# K   Failure case and limitation

Although DreamSteerer achieves high-fidelity editing results, its performance is still limited by the baseline model. For example, as shown in Fig. 23, the DreamBooth model trained on the concept $[duck\_toy]$ lacks subject appearance invariance preservation. With DreamSteerer, although improved source image alignment is achieved, the editing result loses part of the subject appearance information. We emphasize the importance of devising reliable T2I personalization pipelines with improved editability and sample fidelity for future works.

# L   User study

We ask 10 participants to conduct a user preference study of DreamSteerer against different personalization baselines. For each baseline with or without DreamSteerer, a user was asked to evaluate

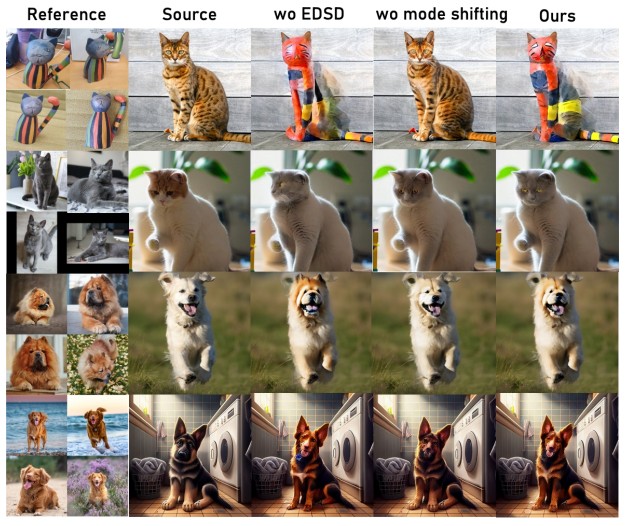

(a) Baseline model: Textual Inversion.

(b) Baseline model: DreamBooth.

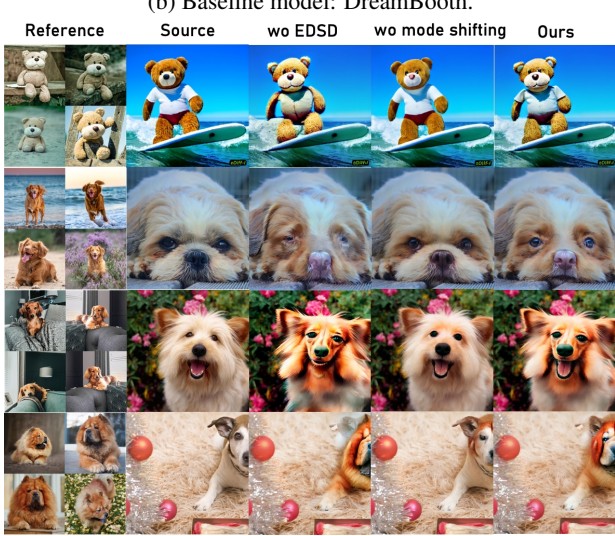

(c) Baseline model: Custom Diffusion.

Figure 13: More ablation study.

Reference    Source    Textual Inversion    **Ours**

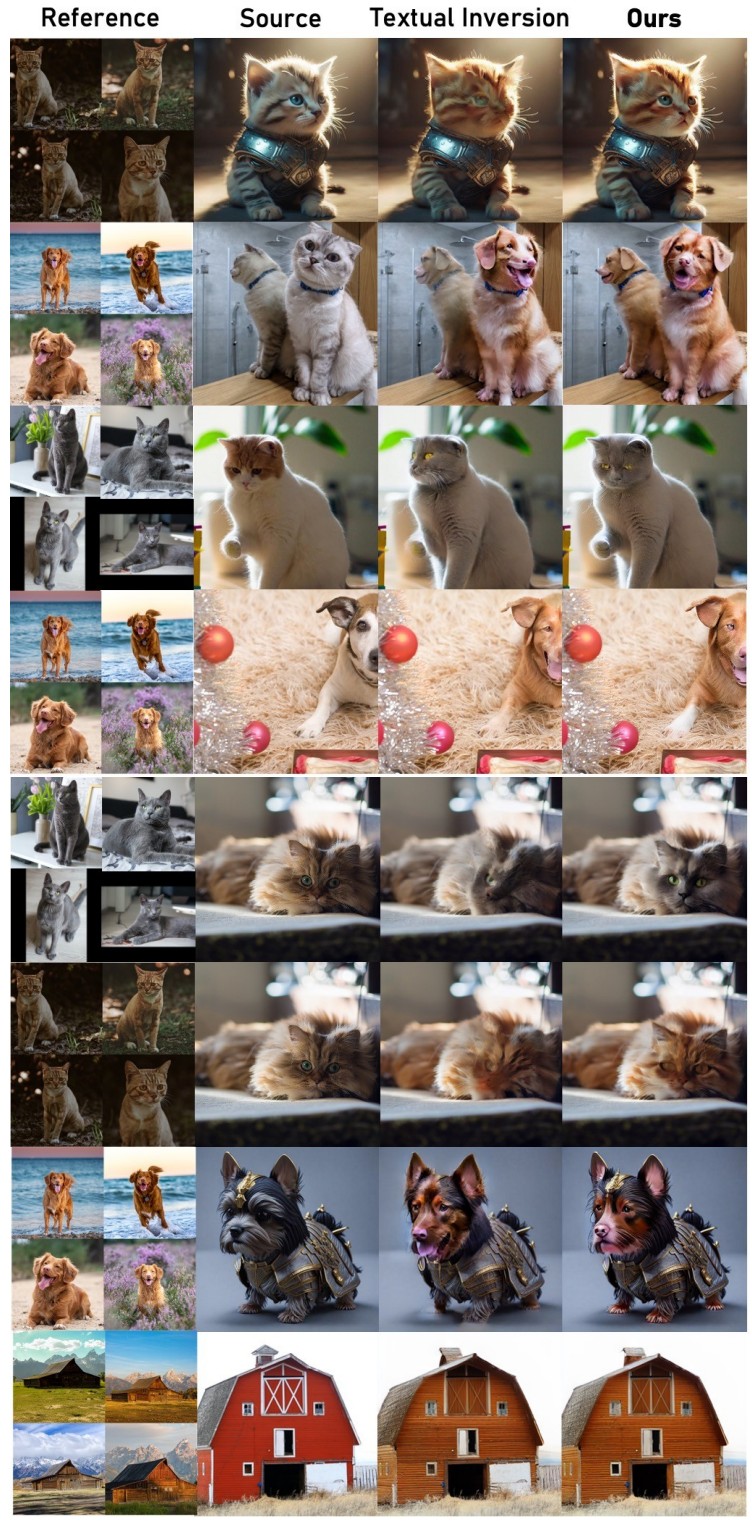

Figure 14: More comparison with Textual Inversion

Reference   Source  Textual Inversion  **Ours**

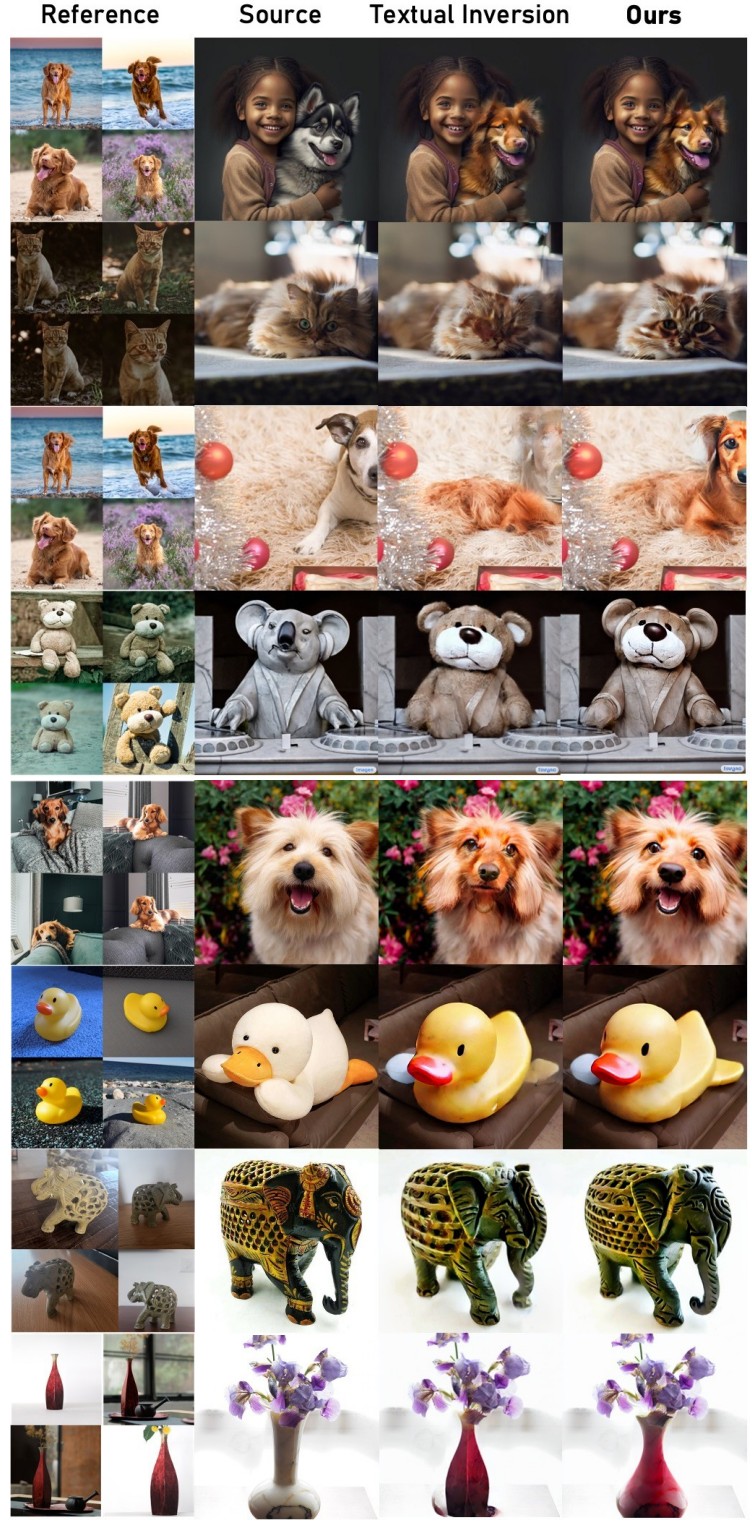

Figure 15: More comparison with DreamBooth

Reference Source Custom Diffusion Ours

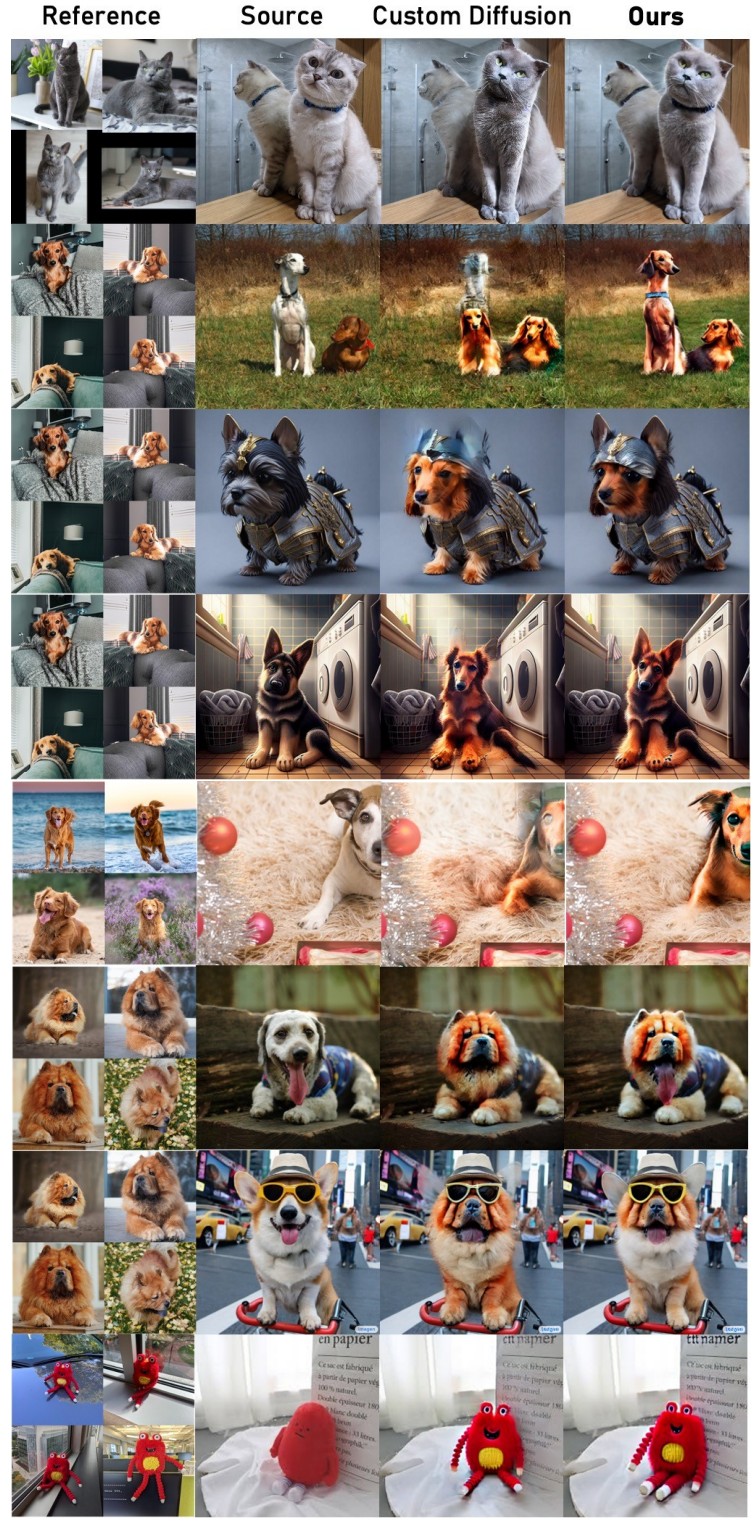

Figure 16: More comparison with Custom Diffusion

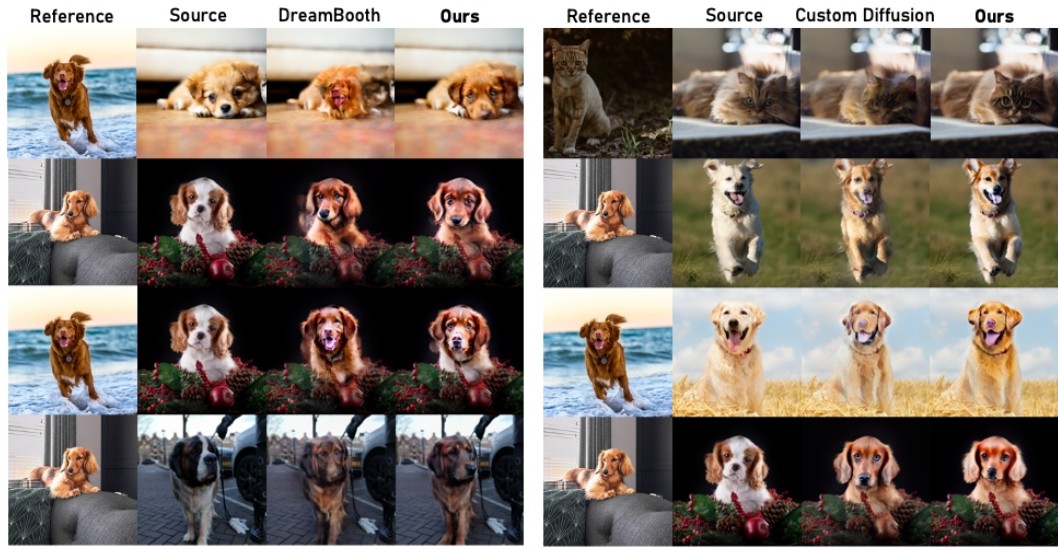

Figure 17: More comparison of one-shot performance with the DreamBooth baseline and Custom Diffusion baseline.

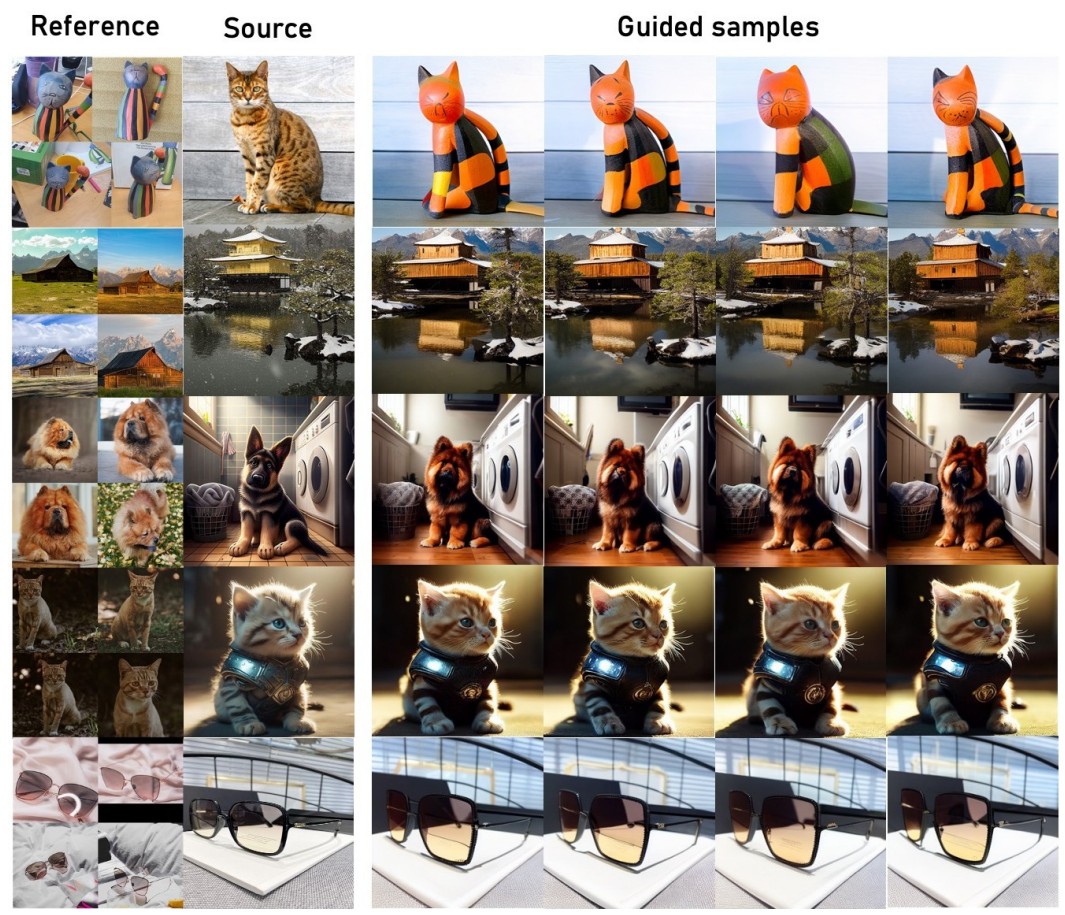

Figure 18: More spatial feature guided sampling results with Textual Inversion baseline model.

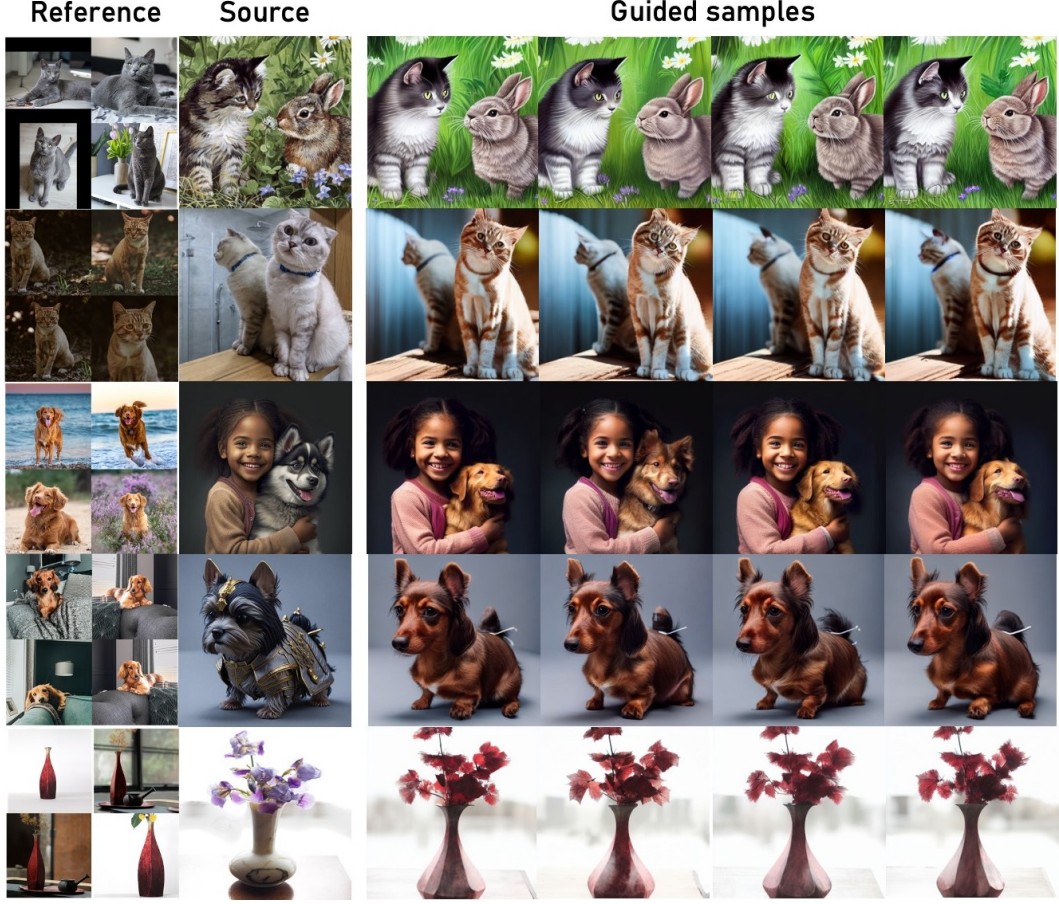

Figure 19: More spatial feature guided sampling results with DreamBooth baseline model.

35 random examples. The examples were given in random order. For each example, the user was asked to rate the overall quality of the edited image on a 1-5 scale (higher is better) with respect to the following criteria:

1. How well does the right image preserve the structural information and the background information of the left image? Consider the accuracy in subject part alignment between two images.

2. How well does the edited subject in the right image preserve the appearance of the subject in the reference images above it?

3. How well is the overall realism and quality of the right image?

A screenshot of the user study page is shown in Fig. 24. As shown in Table. 5, DreamSteerer is preferred by users with a significant margin. By employing the Stable-Diffusion-v-1-4 safety modules, we have prevented the inclusion of NSFW content that could potentially harm the participants.

## M  Broader Impacts

Our research on personalized image editing, particularly in fine-tuning the model to understand personal concepts and applying it to image editing, has significant potential for broad societal impacts. By enabling individuals to seamlessly integrate their personal concepts, styles, and cultural backgrounds into their visual content, our approach democratizes the creative process in digital media. This accessibility enables users across diverse demographics, including artists, content creators, educators, and individuals with limited artistic expertise, to express themselves authentically through content creation. Moreover, by facilitating the creation of personalized content, our method

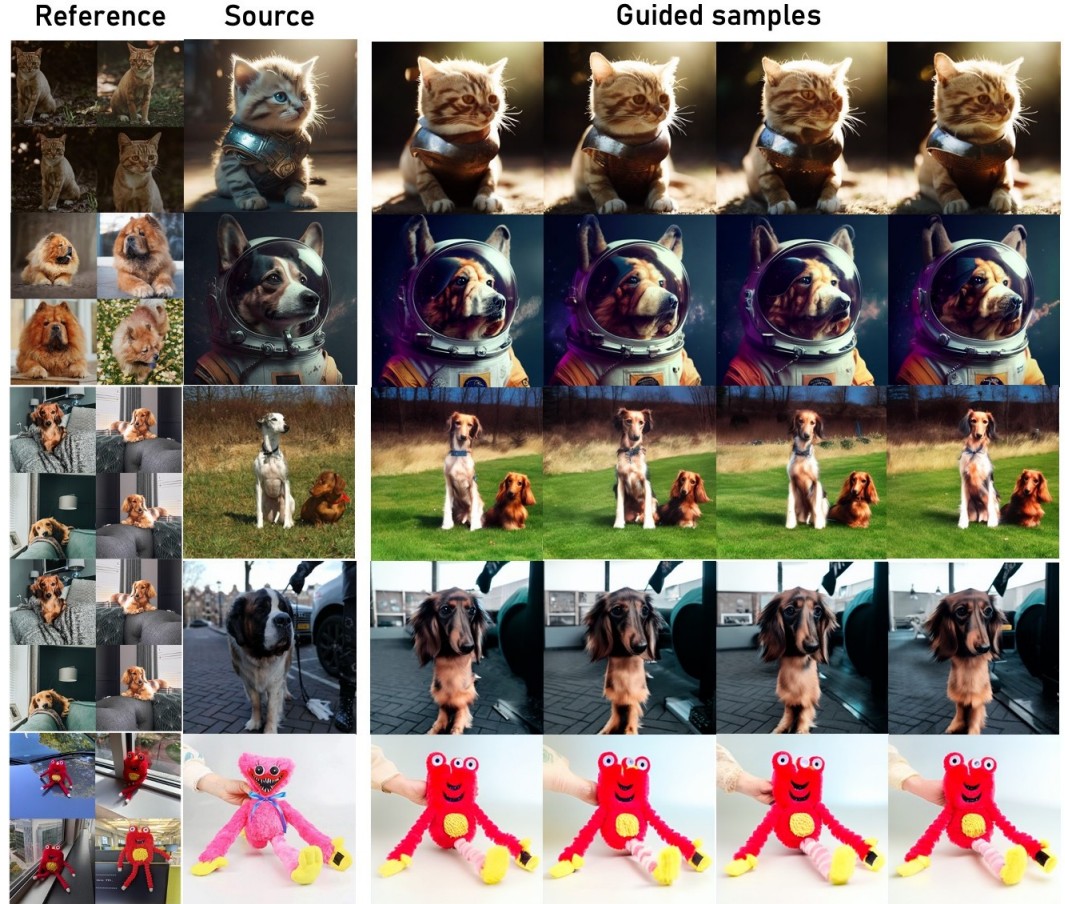

Figure 20: More spatial feature guided sampling results with Custom Diffusion baseline model.

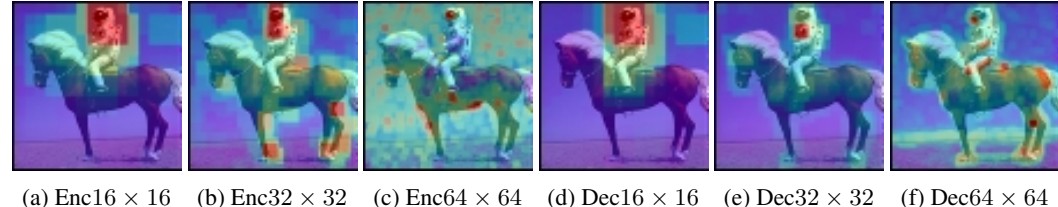

(a) Enc$16 \times 16$   (b) Enc$32 \times 32$   (c) Enc$64 \times 64$   (d) Dec$16 \times 16$   (e) Dec$32 \times 32$   (f) Dec$64 \times 64$

Figure 22: Visualization on averaged cross-attention maps of the DPM UNet encoder and decoder at different resolutions corresponding to 'astronaut' token in the prompt 'an astonaut riding a horse'.

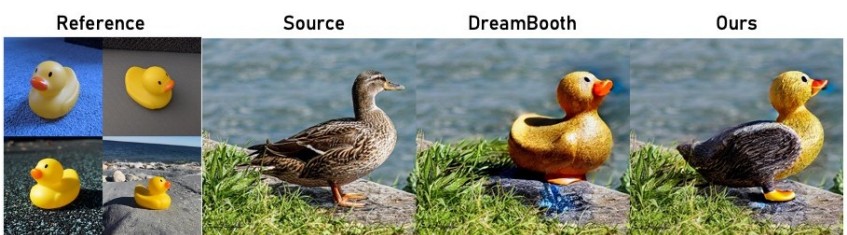

Figure 23: Failure case of DreamSteerer.

This is the set of reference images

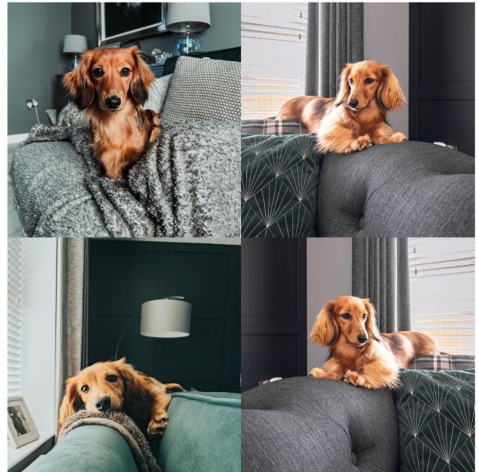

You will be shown a comparison of the source and edited image.

**Please read the instructions below carefully and you will be asked to rate the edited photo based on these criteria later on.**

1. How well does the edited (right) image preserve the structural information and the background information of the source (left) image? Consider the accuracy in subject part alignment between two images.

2. How well does the edited subject in the edited (right) image preserve the appearance of the subject in the reference images above?

3. How well is the overall realism and quality of the edited (right) image?

Here are the source (left) and edited photo (right)

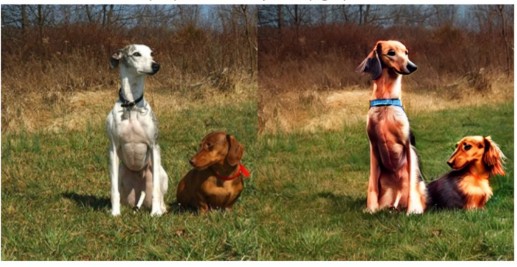

Based on those criteria, please rate the right photo on a scale of 1 to 5 (higher means better).

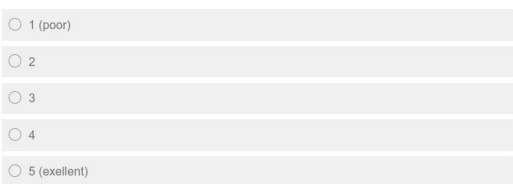

Figure 24: A screenshot of the user study page.

Table 5: User preference study on a 1-5 scale (DreamSteerer uses the same model as baseline).

| User preference (↑) | |
| --- | --- |
| Textual Inversion [16] | 2.48 |
| **DreamSteerer** | **3.78** (+44.5%) |
| DreamBooth [62] | 2.90 |
| **DreamSteerer** | **3.76** (+29.7%) |
| Custom Diffusion [39] | 2.62 |
| **DreamSteerer** | **3.78** (+49.7%) |

has the potential to enhance user engagement and satisfaction in various domains, ranging from social media advertising to educational materials and storytelling. While our research offers a novel approach to personalized content creation, there is a potential for misuse in generating false and harmful content, emphasizing the need for greater caution.

## N   Image attribution

- ViCo dataset: `https://github.com/haoosz/ViCo`
- PIE-Bench: `https://github.com/cure-lab/PnPInversion`
- Textual Inversion datasset: `https://github.com/rinongal/textual_inversion`
- DreamBooth dataset: `https://dreambooth.github.io/`
- Custom Diffusion dataset: `https://www.cs.cmu.edu/~custom-diffusion/`.
- DreamBench: `https://github.com/nousr/dream-bench`
- Cat riding scooter: `https://research.nvidia.com/labs/dir/eDiff-I/`
- Teddybear: `https://research.nvidia.com/labs/dir/eDiff-I/`
- A highly detailed zoomed-in digital painting of a cat dressed as a witch wearing a wizard hat in a haunted house, artstation: `https://research.nvidia.com/labs/dir/eDiff-I/`
- a photo of a corgi dog riding a bike in times square:`https://imagen.research.google/`
- a photo of a cute corgi lives in a house made out of sushi: `https://imagen.research.google`

