# OpenReview forum: "DreamSteerer: Enhancing Source Image Conditioned Editability using Personalized Diffusion Models"
_NeurIPS.cc/2024/Conference — NeurIPS 2024 poster_

### Official Review · Reviewer_kb7M · 2024-07-09

**Soundness:** 3
**Presentation:** 2
**Contribution:** 2
**Rating:** 5
**Confidence:** 5

**Summary:**

This paper proposes DreamSteerer, a plug-in method that enhances existing text-to-image personalization techniques by improving the editability of source images.
 DreamSteerer proposes a novel Editability Driven Score Distillation (EDSD) objective to improve the structural alignment between the source and edited image by performing score distillation with respect to personalized model parameters.
It identifies and addresses mode-trapping issues in EDSD using regularization with spatial feature-guided sampling. The proposed method enhances the editability of several T2I personalization methods.
Moreover, they introduce two key modifications to the Delta Denoising Score framework to enable high-fidelity local editing with personalized concepts.

**Strengths:**

* The problem of enhancing the editability of personalized image editing conditioned on source images is significant and underexplored. This idea is original since it modifies the score distillation methods to better align with the personalized image editing task. They provide an easy-to-use model-agnostic method for any type of personalization.
* The quantitative experiment results are solid and demonstrate the method's effectiveness over the three baselines.
* The presentation and writing are reasonably clear and easy to follow.

**Weaknesses:**

* The method lacks comparisons with editing baselines outside of personalization. It also lacks qualitative or quantitative comparison to the cited preliminary works [8, 14]. For the images that also appear in the Custom-Edit [8] such as rows 1,2,5 in Figure 1 of this paper, the result images in the Custom-Edit [8] paper appear to have much better quality.
* The  "mode shifting regularization" approach is not convincing on the effectiveness of the proposed(Figure 5). In Fig.5, the claimed improvement in editing fidelity from Fig.5 (c) to Fig. 5 (d) does not seem obvious. (also  between "(f)" and "(g)" )
* The paper is missing some important metrics for evaluation. In Section 5, the paper finds that automatic metrics do not fully reflect the superior performance of the proposed method. There is no image quality assessment (IQA) metrics used such as [ref1, ref2, ref3]. Also, the structural similarity metrics globally compare the edited image to the source image, which is unsuitable in this scenario since you want to maintain the similarity in the unedited background region, and a local similarity comparison is better suited. [Please check Questions]
* The method is claimed to be efficient in the abstract section. However, the proposed method's speed and cost are not analyzed or numerically measured. The method appears to require significant computation for a single edit. It is recommended that the computational overhead be reported compared to base personalization methods like DreamBooth.
* The example images in the paper have very low resolution, especially when zoomed in.

**Questions:**

* In the Method section, how is the given personalized model trained, and on which data?
* In Figure 2, what are the reference images? Are they the same as those in Figure 3?
* In Table 1, what do the user preference numbers indicate?
* What is the computational overhead compared to a base personalization method like DreamBooth? How many UNet calls are required to edit one single image?
* How would the performance of the proposed method compare with the recent personalization methods like DreamMatcher?

* The paper is missing some important recent works in personalization:
  * DreamMatcher: Appearance Matching Self-Attention for Semantically-Consistent Text-to-Image Personalization (CVPR 2024)
  * Style Aligned Image Generation via Shared Attention (CVPR 2024)
  * Visual Instruction Inversion: Image Editing via Visual Prompting (NeurIPS 2023)
  * MagicFusion: Boosting Text-to-Image Generation Performance by Fusing Diffusion Models

* References for Weakness section:
  * [ref1] C. Chen, J. Mo, J. Hou, H. Wu, L. Liao, W. Sun, Q. Yan, and W. Lin. Topiq: A top-down approach from semantics to distortions for image quality assessment. IEEE Transactions on Image Processing, 2024.
  * [ref2] J. Ke, Q. Wang, Y. Wang, P. Milanfar, and F. Yang. Musiq: Multi-scale image quality transformer. In Proceedings of the IEEE/CVF international conference on computer vision, 2021.
  * [ref3] W. Zhang, G. Zhai, Y. Wei, X. Yang, and K. Ma. Blind image quality assessment via vision-language correspondence: A multitask learning perspective. In IEEE Conference on Computer Vision and Pattern Recognition, 2023.

**Limitations:**

Discussed in Appendix J

---

> ### Author Rebuttal · Authors · 2024-08-06
>
> Thank you for your constructive feedbacks. Following are responses to your questions.
> >**W1 Further comparison study**
>
> We are targeting image editing with personalized concepts (not exist in vocabulary); thus, editing baselines outside of personalization are not applicable for our task.
>
> **Comparison with Photoswap[1]**: Please refer to general comment part 1 for a setting-level difference and comparison of our work with subject swapping works like Photoswap[1].
>
> **Comparison with Custom-Edit[2]**: In Fig.1, only row 5 uses the same images and the performance is comparable with the same base personalized model, Custom Diffusion. The others (row 1 and 2) either have different source or reference images. Meanwhile, for the cat-statue example in the ablation study shown in Fig.6, we use DreamBooth, which differs from Custom-Edit. For a fairer comparison, in Fig. B (lower) of the attached PDF, we use the same base model, Custom Diffusion, to compare with Custom-Edit. The result of Custom-Edit is more similar to a subject swapping (please refer to general comment part 1), lacking in structural preservation compared to our method (e.g., missing helmet, scarf and incorrect facial structure, etc.). Please refer to general comment part 2 for more comparisons with Custom-Edit and further discussions.
>
> >**W2 Effectiveness in Fig.5**
>
> To emphasize the difference, we provide a higher-resolution version of Fig.5 in Fig.A of the attached pdf. Without the Mode Shifting term, the images in Fig.5f tend to lose a portion of appearance fidelity compared to Fig.5e, resulting in a hybrid appearance closer to the source image (tends to be more silver rather than brown). Conversely, with the inclusion of the Mode Shifting term, the images in Fig.5g maintain appearance fidelity comparable to Fig.5e while displaying patterns akin to the source image, such as the blue collar and the presence of "two cats." This shows the effectiveness of the Mode Shifting term in steering the model to enhance editability for the source image without compromising the personalized subject's appearance information.
>
> A similar observation can be obtained by comparing Fig.5c and Fig.5d, which depict editing outcomes derived from the same prompt. Without the mode shifting term, Fig.5c adheres to the structure of the source image yet displays inaccurate appearance, such as the incorrect coloration of the cat's face.
>
> >**W3 IQA metrics & globally computed structural metrics**
>
> **IQA metrics:** We have conducted user study, which is the mainstream way to assess edited image quality. The evaluation is enriched by using the mentioned Topiq, Musiq and LIQE with [3]. We employed the No-Reference (NR) versions of these metrics, as they are better aligned with the requirements of image editing.
>
> |                        | Topiq | Musiq  | LIQE  |
> |------------------------|-------|--------|-------|
> | Textual Inversion      |  .577 | 68.391 | 4.234 |
> | Textual Inversion+ours |  **.593** | **70.064** | **4.390** |
> | DreamBooth             |  .604 | 70.432 | 4.350 |
> | DreamBooth+ours        |  **.608** | **71.857** | **4.441** |
> | Custom Diffusion       |  .591 | 69.760 | 4.240 |
> | Custom Diffusion+ours  |  **.612** | **71.450** | **4.414** |
>
> Our work consistently improves the overall image quality of the baselines.
>
> **Structural similarity metrics computed globally:** Maintaining a structural similarity for the unedited regions are also important as the editing may modify subject-irrelevant regions, which is undesired.
>
> >**W4 Computational overhead**
>
> Please refer the general comment part 3.
>
> >**W5 Low resolution of images**
>
> To ensure a smooth submission process, we reduced the size of the images, which resulted in a lower resolution. To ensure that our work is presented with clarity and detail necessary for thorough evaluation and understanding, we will replace all figures with their original, high-resolution versions for camera ready version (similar to the image quality in the attached pdf).
>
> >**Q1 Training and data of personalized model**
>
> We conduct evaluation based on publicly accessible checkpoints provided by the open repository[4]. These checkpoints, specifically for Textual Inversion, DreamBooth, and Custom-Diffusion, have been trained on 16 identical concepts from the ViCo dataset. They exhibit generation fidelity that aligns with the results reported in the original papers, which ensure a fair and direct comparison across different personalization methods.
>
> >**Q2 Reference images in Fig.2**
>
> Apologies for the oversight regarding the reference images in Fig.2. Indeed, they are identical to those presented in Fig.3. We will ensure that Fig.2 is updated accordingly in the camera-ready version.
>
> >**Q3 User preference numbers**
>
> These ratings assess the overall quality of the edited images where a higher score indicates better quality. The rating criteria include (1) structure and background preservation from the source image, (2) appearance and concept preservation from the reference images, (3) overall realism and quality of the edited image. Please refer to appendix K of the main paper for details.
>
> >**Q4 Computational overhead compared to a base method**
>
> Please refer to the general comment part 3.
>
> >**Q5 Compare with DreamMatcher**
>
> DreamMatcher is an interesting concurrent work. However, its focus is on boosting generation fidelity rather than editing, which is the core of our work. Due to this fundamental difference in purpose, it is not directly comparable to our work. We have acknowledged this work in our extended literature review.
>
> >**Q6 Missing recent works**
>
> We have added an extended literature review on T2I personalization, where the mentioned works are discussed (response to reviewer zDhi W1).
>
> [1] Photoswap: Personalized subject swapping in images
>
> [2] Custom-edit: Text-guided image editing with customized diffusion models
>
> [3] https://github.com/chaofengc/IQA-PyTorch
>
> [4] https://github.com/KU-CVLAB/DreamMatcher

---

> > ### Comment · Reviewer_kb7M · 2024-08-13
> >
> > Thank you for your reply.
> >
> > The concerns were addressed and I am increasing my score.

---

> > > ### Author Response · Authors · 2024-08-13
> > >
> > > Thank you for your decision to increase the score and your dedicated effort in reviewing our paper! Your comments helped clarify and improve our work.

---

### Official Review · Reviewer_2C3N · 2024-07-12

**Soundness:** 3
**Presentation:** 3
**Contribution:** 2
**Rating:** 4
**Confidence:** 4

**Summary:**

The paper introduces DreamSteerer, a method to enhance the editability of source images using personalized diffusion models in text-to-image (T2I) personalization. Existing methods often fail to maintain edit fidelity when applied to new contexts due to limited reference images and adaptability issues. DreamSteerer addresses this by introducing the Editability Driven Score Distillation (EDSD) objective, which enhances image editability. To mitigate a mode trapping issue in EDSD, the paper proposes mode shifting regularization with spatial feature guided sampling. This technique aligns the model more closely with the source image structure while preserving personalized concept appearances. Extensive experiments show that DreamSteerer significantly improves editability and efficiency across various T2I personalization baselines.

**Strengths:**

1、The method is designed as a plug-in compatible with arbitrary personalization baselines.  This versatility enhances its applicability across various models and use cases
2、DreamSteerer requires only a small number of fine-tuning steps (~10) to achieve significant improvements, making it computationally efficient.

**Weaknesses:**

1、While the paper presents a novel approach, it would benefit from a more thorough discussion regarding its innovative aspects in comparison to existing work in Object Driven Image Editing, such as DreamEdit[1] and SWAPANYTHING[2]. It appears that similar mechanisms are already in place within these frameworks, and a detailed exploration of how the current method distinguishes itself or builds upon these concepts would be valuable.
2、A broader experimental evaluation is suggested.  This should include comparative studies with existing image editing works and other Object Driven Image Editing techniques mentioned in [1,2].
3、The proposed method relies on certain heuristics, such as spatial feature guided sampling and the choice of specific parameters for mode shifting regularization.  These heuristics may not generalize well across different datasets or tasks, leading to suboptimal performance in some scenarios.

[1] DreamEdit: Subject-driven Image Editing
[2] SWAPANYTHING: Enabling Arbitrary Object Swapping in Personalized Visual Editing

**Questions:**

Figure 14 and 13 is same.
Insufficient resolution of figures provided in the paper

**Limitations:**

The paper discussed the limitations of this work.

---

> ### Author Rebuttal · Authors · 2024-08-06
>
> Thank you for the valuable feedbacks. Following are responses to your questions.
>
> > **W1 Comparison to Object Driven Image Editing works**
>
> We thank the reviewer for the constructive suggestion and acknowledgement on the novelty of our method. For a detailed discussion on how our approach differs from previous works in subject swapping, including a comparison with DreamEdit[1], please refer to general comment part 1. SwapAnything[3] is also an interesting concurrent work follows a similar setting as DreamEdit, which is different from the setting of our work. We will defer detailed comparison of SwapAnything to future work once they release their codes.
>
> > **W2 Broader experimental evaluation**
>
> **Comparison with different subject swapping work**: Please refer to general comment part 1 and response to W1.
>
> **Comparison using different image editing method**:
> Please refer to general comment part 2.
>
> >**W3 Dependence on heuristics**
>
> We thank the reviewer for mentioning the concern. Our method does not require a careful tuning on the hyper-parameters. We use the same learning rates as the original personalization baselines. Beyond this, the same set of hyper-parameters are shared among the personalization baselines, irrespective of the concept or source image involved.
>
> > **Q1 Figure 14 and 13 is same. Insufficient resolution of figures provided in the paper**
>
> * Thanks for noticing the duplicated image, we will avoid this for the camera-ready version.
> * We apologize for the insufficient resolution of images in our main paper. To avoid potential network issues and ensure a smooth submission process, we reduced the size of the images, which resulted in a lower resolution. To ensure that our work is presented with clarity and detail necessary for thorough evaluation and understanding, we will replace all figures with their original, high-resolution versions for camera-ready version.
>
> [1] DreamEdit: Subject-driven Image Editing
>
> [2] Custom-edit: Text-guided image editing with customized diffusion models
>
> [3] Swapanything: Enabling arbitrary object swapping in personalized visual editing

---

> > ### Comment · Area_Chair_Hnku · 2024-08-13
> >
> > Reviewer 2C3N, do you have any additional questions or feedback?

---

> > ### Comment · Reviewer_2C3N · 2024-08-14
> > **comment from reviewer**
> >
> > Thank you for answering my questions. After reading the author's rebuttal, I think the current version still needs revision in terms of (I) experiments on the sensitivity of the heuristic network to hyperparameters (the specific value of the hyperparameter λ is also missing in the implementation details), (ii) a more significant performance improvement compared to Custom-Edit, and (iii) further revisions to the current paper, such as supplementing the paper with additional details like λ and also the use of lossless compression to reduce the volume while maintaining the sharpness of the image. Therefore, I will keep my rating same.

---

> > > ### Author Response · Authors · 2024-08-14
> > >
> > > Thanks for your suggestions. Following are responses to your concerns.
> > >
> > > >**Sensitivity of hyper-parameter $\lambda$**
> > >
> > > In all our experiments, regardless of the personalization baseline used, we set $\lambda$ to 15. Figures 15-17 in the appendix illustrate that this parameter choice reliably produces guided samples that not only retain the structural layout of the source image but also ensure the appearance fidelity of the reference subject is preserved. This level of performance is consistent across a wide variety of personalization baselines, source images, and reference images. Our approach eliminates the need for users to meticulously adjust hyperparameters to achieve significant improvements in editability. A comprehensive ablation study on $\lambda$ will be detailed in the appendix of the camera-ready manuscript.
> > >
> > > >**Improvement compared to Custom-Edit**
> > >
> > > The improvement compared to Custom-Edit is less significant for metrics related to the alignment with source image such as SSIM and LPIPS. This is because Custom-Edit uses Prompt-to-Prompt as the base editing model which enforces the source structural alignment with the attention injection mechanism. However, when our method is applied as a plug-in, we consistently see enhanced performance in reference appearance alignment metrics (CLIP score) and overall image quality (Topiq, Musiq, and LIQE), as illustrated in Fig.B.
> > >
> > > Additionally, as noted in part 2 of the general comments, the Prompt-to-Prompt model used by Custom-Edit fails to produce valid edits for DreamBooth due to its reliance on a latent-state inversion process with Null-Text Inversion, which lacks editability for certain personalized models. Unlike Prompt-to-Prompt, the DDS-SM base editing method used in our main paper does not encounter this issue and is effective across various personalization approaches. Our approach, using DDS-SM as the base editing method, is not limited to a specific personalization technique; instead, it offers a versatile plug-in solution that enhances performance across a broad range of metrics for different personalization baselines.
> > >
> > > >**Lossless compression**
> > >
> > > Thanks for the suggestion. We will consider this for camera-ready version of the manuscript and will ensure that the presented figures have sufficient resolution that matches the quality of images in the attached pdf file.
> > >
> > > Thanks again for your valuable suggestions. We sincerely hope you can consider re-scoring our paper.

---

### Official Review · Reviewer_zDhi · 2024-07-12

**Soundness:** 3
**Presentation:** 3
**Contribution:** 3
**Rating:** 5
**Confidence:** 4

**Summary:**

Aiming at addressing unsatisfactory editability on the source image, this paper proposes a novel plug-in method for augmenting existing T2I personalization methods.  Specifically, this framework finetunes the personalization parameters by training a novel Editability Driven Score Distillation objective under the constraint of a Mode Shifting regularization term based on spatial feature-guided samples. Extensive experiments validate the effectiveness of the proposed method.

**Strengths:**

1. The proposed method can be applied current T2I personalization models to perform custom editing as a plug-in.

2. Efficiently, the framework  DreamSteerer requires only a small number of fine-tuning steps (∼ 10) to achieve significant improvement in editability on a source image.

**Weaknesses:**

1. In related work, the authors lack a discussion of recent works for T2I personalization. Most of the works are in the first half of last year.

2. This framework introduces many computing loads in there modules in Fig.3, which worries me about the inference efficiency of this method. It is suggested that the authors can provide some explanation about computational complexity in experiment section.

3. There are some excellent methods for custom editing, such as [1] [2]. It is recommended to compared with these strong baseline to show the effectiveness of this method.

4. As for qualitative results in this paper, obviously, the proposed method forces consistency of the object shape between the source object and target object, resulting in the destruction of the edited object identity compared the reference. This may not be the editing result that the user wants. It is suggested the authors can provide analysis for this.

[1] Custom-edit: Text-guided image editing with customized diffusion models.

[2] DreamEdit: Subject-driven Image Editing.

**Questions:**

See weakness.

**Limitations:**

Although DreamSteerer achieves high-fidelity editing results, its performance is still limited by the baseline model.

---

> ### Author Rebuttal · Authors · 2024-08-06
>
> Thank you for your constructive comments and questions. Following are responses to your concerns.
>
> > **W1 Insufficient related work of T2I personalization**
>
> Thank you for the suggestion. We provide an additional literature review related to T2I personalization mostly from this year below:
>
> Recent studies in encoder-based personalization[1-4] propose new training paradigms to condition a Diffsuion Model on single or multiple input images. Although these works can enable faster inference, they necessitate extensive pre-training and typically limit application to particular domains. New forms of conditionings, objectives or adaptors are proposed for different purposes such as stylization[5-7], improved identity preservation[8,9], composability[10-13], or generation fidelity[14]. Unlike these works, we focus on addressing an inherently different task of improving the editability of personalized Diffusion Models.
>
> We will include these in our revised paper.
>
> [1] JeDi: Joint-Image Diffusion Models for Finetuning-Free Personalized Text-to-Image Generation, CVPR 2024.
>
> [2] Instantbooth: Personalized text-to-image generation without test-time finetuning, CVPR 2024.
>
> [3] RealCustom: Narrowing Real Text Word for Real-Time Open-Domain Text-to-Image Customization, CVPR 2024.
>
> [4] Hyperdreambooth: Hypernetworks for fast personalization of text-to-image models, CVPR 2024.
>
> [5] Customizing Text-to-Image Models with a Single Image Pair, arXiv 2024.
>
> [6] Visual instruction inversion: Image editing via image prompting, NeurIPS 2023.
>
> [7] Style aligned image generation via shared attention, CVPR 2024.
>
> [8] When stylegan meets stable diffusion: a w+ adapter for personalized image generation, CVPR 2024.
>
> [9] IDAdapter: Learning Mixed Features for Tuning-Free Personalization of Text-to-Image Models, CVPR 2024.
>
> [10] Attention Calibration for Disentangled Text-to-Image Personalization, CVPR 2024.
>
> [11] Orthogonal adaptation for modular customization of diffusion models, CVPR 2024.
>
> [12] Omg: Occlusion-friendly personalized multi-concept generation in diffusion models, ECCV 2024.
>
> [13] Magicfusion: Boosting text-to-image generation performance by fusing diffusion models, ICCV 2023.
>
> [14] Dreammatcher: Appearance matching self-attention for semantically-consistent text-to-image personalization, CVPR 2024.
>
> > **W2 Computational complexity**
>
> Please refer to general comment part3.
>
> > **W3 Comparison with existing methods**
>
> **Comparison with DreamEdit[2]**:
> Please refer to general comment part 1 for a discussion on how our work differ from subject-driven editing works like DreamEdit[2]. As illustrated in Fig.C of the attached rebuttal pdf, compared with our work, DreamEdit often exhibits severe distortion or failure in maintaining structural consistency with the source image.
>
> **Comparison with Custom-Edit[1]**: Please refer to general comment part 2.
>
> > **W4 Forcing shape consistency with source**
>
> * Please refer to general comment part 1 for a clarification on how our work connects with rigid text-driven image editing and differs from subject swapping.
> * Here we also illustrate several advantages that the feature of aligning with the source structure might offer for real-world applications:
>     * Enhanced creative experimentation: content creators have the freedom to explore different appearances based on different reference concepts without compromising the essence of the source image. This facilitates the exploration of diverse aesthetic outcomes with ease.
>     * Consistency and recognizability: in branding and media content creation, maintaining the original structure ensures that if viewers are familiar with the original content, it enhances brand recognition and ensures continuity of visual content.
>     * Contextual integrity preservation: in fields such as cultural heritage, aligning edits with the source structure can be crucial for maintaining the context and authenticity of visual information.
>
> [1] Custom-edit: Text-guided image editing with customized diffusion models
>
> [2] DreamEdit: Subject-driven Image Editing

---

> > ### Comment · Area_Chair_Hnku · 2024-08-13
> >
> > Reviewer zDhi, do you have any additional questions or feedback?

---

> > ### Comment · Reviewer_zDhi · 2024-08-14
> > **Official Comment by Reviewer zDhi**
> >
> > Thank you for your detailed reply. It addresses most of my concerns.
> >
> > I will keep my rating to borderline accept.

---

> > > ### Author Response · Authors · 2024-08-14
> > >
> > > Thank you for your dedication and valuable input! We also appreciate your decision to maintain a borderline accept rating. Your suggestions have helped to improve and clarify our work.

---

### Official Review · Reviewer_AnAq · 2024-07-13

**Soundness:** 4
**Presentation:** 4
**Contribution:** 4
**Rating:** 7
**Confidence:** 4

**Summary:**

DreamSteerer is a proposed fine-tuning pipeline for personalized diffusion models, designed to enhance the custom editability of these models. The authors point out that naively incorporating existing image editing and personalization methods—such as employing score distillation and DDS where the differentiable function is initialized by the source image latent space—results in low fidelity and lack of editability relative to the source image. To address these issues, they propose the following additions that comprise DreamSteerer:
1. Editability driven score distillation (EDSD), to increase the editability of the model by distillation information from the source model $\epsilon_{\phi_0}$ to the personalized diffusion model  $\epsilon_{\phi_0}$ through a single-step perturbed source latent state
2. Spatial Feature Guided Sampling and mode shifting regularization to overcome the mode trapping issue introduced by EDSD (commonly observed in score distillation)
3. Source score bias correction and automatic subject masking to address the distribution shift from personalization, and to focus on the relevant aspects of the image during training respectively.

The authors present thorough qualitative and quantitative results across three metrics: semantic similarity, perceptual similarity, and structural similarity, along with a human preference evaluation. They demonstrate that DreamSteerer consistently improves upon existing baseline methods and include ablation studies to justify various components of the pipeline.

**Strengths:**

Originality: The work is well-motivated and addresses a limitation in the intersection of personalization and image editing, which is a relatively underexplored area. Although DDS has been employed for general text-driven editing, the application of DDS to enhancing editability in the personalization setting is a novel approach. In addition to combining the existing ideas, the authors introduce further modifications to improve final results.

Clarity: The paper clearly explains each learning objective with detailed formulations, supported by an overarching diagram. The results are well-organized and articulated, presenting both qualitative and quantitative analyses. This structure provides a cohesive narrative that justifies the design decisions made throughout the work.

Quality: The proposed algorithm introduces a novel approach which improves over existing methods in the personalized image editing domain, and the work presents sufficient ablation studies over core components to reinforce the reliability of the proposed approach.

Significance: The proposed method for personalized image editing with few fine-tuning steps required has potential for widespread adoption. DreamSteerer's ability to maintain structural integrity while enabling personalization edits makes the work applicable across various creative domains, which may demand precise edits.

**Weaknesses:**

The paper’s results would benefit from benchmarking on more extensive personalization datasets, such as CustomConcept101. Currently, the evaluation is based on only 16 concepts from the ViCo repository, a design decision that lacks strong justification. While the selected concepts show a reasonable level of diversity, evaluating the method across a broader set of personalized concepts would enhance the robustness and generalizability of the results.

Additionally, there are minor implementation details that could be clarified in the Experiments section. Specifically, the paper mentions using pre-trained checkpoints provided by ViCo. However, the ViCo GitHub repository does not link to checkpoints for Textual Inversion, DreamBooth, and Custom Diffusion, leading to some confusion about the source of these pre-trained checkpoints. Providing clear information on where these checkpoints were obtained would help improve the reproducibility of the work.

**Questions:**

I have minor questions regarding the implementation details to clarify some confusion:
1. Source of Real-World Images: Can you provide more details on the 70 random real-world images used for editing? Were these images scraped from the web or sourced from an existing dataset? Additionally, were images containing a shared superclass used to enhance the feasibility of the editing tasks?
2. Clarification on Figure 5: In Figure 5, it’s unclear where the results in subfigures (e) to (g) originate from. Are these images based on specific source images, or are they entirely generated without any reference images, from the finetuned personalization model?

**Limitations:**

The authors have adequately addressed the limitations and potential societal impact of their work. Specifically, they describe and visualize the limitations of DreamSteerer, which can be constrained by the baseline personalization model, and suggest ways to avoid fidelity inconsistencies in future similar works.

---

> ### Author Rebuttal · Authors · 2024-08-06
>
> Thank you for the insightful comments and questions. Following are responses for your questions.
>
> > **W1 Benchmarking on more extensive personalization datasets**
>
> Thank you for your feedback. Our decision to evaluate based on 16 concepts from the open repository[1] was driven by the availability of comparable, publicly accessible checkpoints. These checkpoints, specifically for Textual Inversion, DreamBooth, and Custom-Diffusion, are trained on the same concepts within the ViCo dataset and demonstrate generation fidelity consistent with their original publications. This approach allowed us to ensure a fair and direct comparison across different personalization methods. We acknowledge the importance of evaluating our method across a broader set of personalized concepts to enhance the robustness and generalizability of our results. We plan to explore this in future work, aiming to provide a more comprehensive evaluation across additional concepts and personalization baselines.
>
>
> > **W2 Pre-trained checkpoints**
>
> Thank you for pointing out the need for clarification regarding the implementation details. We apologize for any confusion caused by the oversight in our manuscript. The pre-trained checkpoints used in our work were sourced from the DreamMatcher GitHub repository[1], that has been trained on the ViCo dataset. We will update our manuscript to include this accurate source information to enhance the reproducibility of our work.
>
> > **Q1  Source of Real-World Images**
>
> The images we use are collected from 3 main sources:
> * DreamBench[2]
> * PIE-Bench[3]
> * Web data
>
> Our selection criteria were particularly focused on images where the source subject exhibited a significant structural difference from the reference subjects, yet maintained a shared superclass with these references. Such selection criteria enable us to better evaluate the robustness of the personalized editing with significant structural misalignments between source and reference subjects. We will include these details in our updated paper.
>
> > **Q2  Clarification on Figure 5**
>
> Thank you for seeking clarification regarding Figure 5 in our manuscript. We understand the confusion and are happy to provide a more detailed explanation.
>
> These figures are random generations **using different diffusion models** based on the prompt 'A photo of a sks cat standing next to a mirror', where sks corresponds to the personalized concept in Fig.5b. Specifically:
>
> * Fig.5e is generated by a base personalized model trained on the images shown in Figure 5b.
> * Fig.5f results from fine-tuning the base model with only the Editability Driven Score Distillation (EDSD) term.
> * Fig.5g results from fine-tuning the base model with both the EDSD term and the Mode Shifting term.
>
> Our analysis reveals that without the Mode Shifting term, the images in Fig.5f tend to lose a portion of appearance fidelity compared to Fig.5e, resulting in a hybrid appearance closer to the source image. Conversely, with the inclusion of the Mode Shifting term, the images in Fig.5g maintain appearance fidelity comparable to the base model (Fig.5e) while displaying patterns more akin to the source image, such as the blue collar and the presence of "two cats." This demonstrates the effectiveness of the Mode Shifting term in steering the model to enhance editability for the source image without compromising the personalized subject's appearance information. We will include this in our revised paper.
>
> [1] https://github.com/KU-CVLAB/DreamMatcher
>
> [2] https://github.com/nousr/dream-bench
>
> [3] https://github.com/cure-lab/PnPInversion

---

> > ### Comment · Area_Chair_Hnku · 2024-08-13
> >
> > Reviewer AnAq, do you have any additional questions or feedback?

---

> > > ### Comment · Reviewer_AnAq · 2024-08-13
> > >
> > > Thank you authors for the detailed feedback in addressing some of my concerns. I will keep my original score.

---

> ### Author Response · Authors · 2024-08-13
>
> Thank you for your positive feedback and the time you invested in reviewing our work. Your suggestions helped clarify and improve our work.

---

### Author Rebuttal · Authors · 2024-08-06

## Global comment
We thank all reviewers for their great effort and suggestions. Some general clarifications are provided below for better understanding of both our task and our method. We will include these details in our revised paper.
> **Part 1**

***Connection with text-driven image editing***

Traditional text-driven image editing methods generally fall into two categories: rigid editing (e.g., [1,2]) and non-rigid editing like[3]

We conclude the desiderata of rigid editing as follows:
* the overall structure of edited image should align with the source image,
* the edited part should follow the input instruction,
* the instruction-irrelevant part should be preserved as much as possible.

Non-rigid editing focuses on changing the view or pose of a subject in the source image while preserving the background.

In this work, **we consider extending text-driven rigid editing** to editing scenario like 'a photo of a (silver → $v*$) cat', where $v*$ represents a specific concept derived from reference images. This concept is more detailed than generic descriptions (e.g., 'brown'), capturing intricate appearance and semantic information. We provide a flexible plug-in to bridge the gap in editability between specific concept and textual conditionings using Diffusion Models, offering a unique contribution not addressed by existing methods.

***Difference with subject swapping***

Subject swapping methods like DreamEdit[4] and Photoswap[5] diverge from our approach in their main criteria. While these methods prioritize alignment with the source subject's location and pose, they do not necessitate maintaining the original structural details as our method does. Furthermore, these works demand a stricter preservation of subject identity and typically do not require a significant level of concept extrapolation like our work, e.g., from a short cat to a tall cat. In Fig.C of the attached rebuttal pdf, we provide a comparison with DreamEdit and Photoswap in scenarios involving significant structural gaps. Compared with our work, these works often exhibits severe distortion or fails to maintain structural consistency with the source image.

[1] Prompt-to-prompt image editing with cross attention control

[2] Instructpix2pix: Learning to follow image editing instructions

[3] Masactrl: Tuning-free mutual self-attention control for consistent image synthesis and editing

[4] Dreamedit: Subject-driven image editing

[5] Photoswap: Personalized subject swapping in images
>**Part 2**

***Comparison using different image editing method***

* We employ a variant of Delta Denoising Score as the base editing model as this method provides stable editing results with all the personalized models we use.
* However, DreamSteerer is not restricted to a specific type of editing pipeline. We also evaluate the effectiveness of our method as a plug in for Custom-Edit[3], which directly combines Custom Diffusion with Prompt-to-Prompt. The numerical results are as follows:

|                 | CLIP B/32  | CLIP L/14 | LPIPS (Alex) | LPIPS (VGG) | SSIM | MS-SSIM | Topiq | Musiq  | LIQE  |
|-----------------|------------|-----------|--------------|-------------|------|---------|----------|--------|-------|
| Custom-Edit      |       .748 |      .727 |         .141 |        .210 | .793|    .899|     .564 | 67.942 | 3.974 |
| Custom-Edit+ours |       **.750** |      **.729** |         .141 |        **.209** | .793|    .899 |     **.565** | **67.948** | **3.981** |
* As shown in Fig.B (upper) of the attached pdf, Custom-Edit might not adequately preserve the appearance information of the reference subject. Our proposed method can effectively improve the reference subject appearance preservation when implemented as a plug-in, without compromising the structural alignment with the source image.
* Meanwhile, as shown in Fig.D of the attached pdf, combining Prompt-to-Prompt with DreamBooth can introduce significant appearance artifacts in the edits, which is the main reason we did not use it as the base editing method. Prompt-to-Prompt relies on a source latent state inversion process, typically through Null-Text Inversion (NTI). However, parameter update during personalization can shift the model distribution for the source class, compromising the editability of the inverted latent state chain with NTI. In comparison, the Delta Denoising Score based edited method employed in our work does not require a inversion process, providing more robust performance across different types of personalization baselines. We believe this phenomenon worth further investigation and encourage future works to develop new inversion techniques specifically tailored for the personalized models.

>**Part 3**

***The computational cost of DreamSteerer***

The total sampling & fine-tuning time of DreamSteerer takes ~2 minutes on a single V100 GPU, that comes from two parts: (1) spatial feature guided sampling: this part is tuning-free and contains a DDIM inversion process on the single source image and a guided generation process. With a DDIM scheduler step size of 50, this process takes ~1 minute on a single v100 GPU, (2) the EDSD fine-tuning on personalized model parameters: this process takes a small number of 10 optimization steps with a batch size of 1 and a gradient accumulation step of 10, the fine-tuning time depends on the exact base personalized model. For the largest model DreamBooth, which uses full fine-tuning, it takes ~1 minute. We provide a comparison of the total number of UNet prediction steps (NFE) between our fine-tuning strategy and the baseline personalized models.
|                   | NFE  |
|-------------------|------|
| Textual Inversion | 5000 |
| DreamBooth        | 1600 |
| Custom Diffusion  | 2000 |
| Ours              | 200  |

The fine-tuning and inference of the whole pipeline is also able to be conducted on a single 3090 GPU for the largest base model DreamBooth.

---

> ### Author Response · Authors · 2024-08-13
> **[Gentle reminder] One day remains before the end of discussion period**
>
> Dear reviewers and AC,
>
> Thank you for the time and effort you have invested in reviewing our manuscript and providing insightful feedback. We have carefully considered each of your comments and have prepared detailed responses to address your questions and concerns. It is our hope that these responses have satisfactorily resolved your concerns. As we near the end of the discussion period, we welcome any additional questions or feedback you may have.
>
> Warm regards,
>
> Authors

---

### Decision · Program_Chairs · 2024-09-25

**Decision:**

Accept (poster)

**Comment:**

Though the paper has mixed reviews, the majority of reviewers are leaning toward acceptance. The reviewers appreciate that DreamSteerer enhances the users ability to edit personalized diffusion models, and that the authors have conducted a thorough quantitative and qualitative analysis of the results, including a human evaluation of results. The AC team agrees that the value of the contribution is sufficient for publication at NeurIPS and thus recommend acceptance.